# CoPE: A Framework for Optimizing Coordination between Planning and Execution in LLM-based Agents

Huanxi Liu [1 2 3]   Kun Hu [1]   Qiang Wang [1 2 3]   Yuanzhao Zhai [1 2 3]   Feng Dawei [1 2 3]   Bo Ding [1 2 3]
Huaimin Wang [1 2 3]

## Abstract

Fine-tuning Large Language Models (LLMs) as autonomous agents on domain-specific data has emerged as a promising paradigm for tackling interactive, real-world tasks. However, existing studies have overlooked the critical coordination between long-term planning and multi-step execution in optimizing agent capabilities. This oversight leads to the propagation of impractical plans and plan-deviated trajectories within the optimization process, resulting in suboptimal task performance and hindering the further development of LLM-based agents in long-horizon tasks. To bridge this gap, we propose **CoPE**, a novel framework that explicitly integrates planning–execution coordination into LLM-based agent optimization. CoPE employs Self-Refining MCTS to generate task plans and multiple execution trajectories through environment interactions. By quantifying the coordination between planning and execution, CoPE assigns higher optimization weights to well-coordinated samples, enabling LLM-based agents to learn better planning and execution policies. Extensive experiments demonstrate that CoPE substantially improves agent coordination, outperforming state-of-the-art baselines on benchmarks comprising two long-horizon multi-step tasks. Codes and data are available at https://github.com/Octobrist/CoPE.

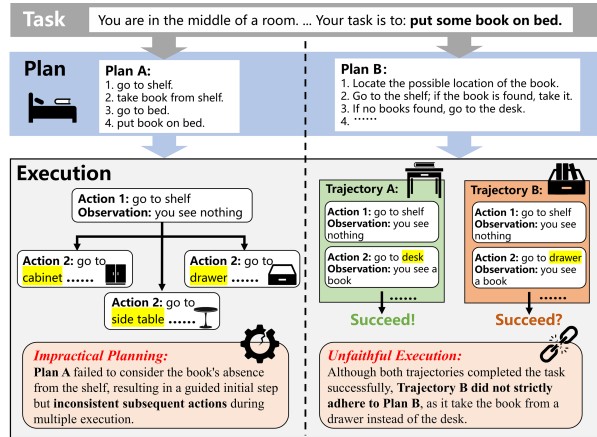

*Figure 1.* Two examples of incoordination between LLM-based agent planning and execution in AlfWorld (Shridhar et al., 2021). The left panel shows how impractical planning leads to inconsistent execution. The right panel illustrates unfaithful execution, where *Trajectory B* deviates from *Plan B* but still succeeds, making it hard to distinguish from *Trajectory A* solely via task rewards.

## 1. Introduction

Large Language Models (LLMs) (Achiam et al., 2023; Liu et al., 2024; Yang et al., 2025) have demonstrated remarkable capabilities in context comprehension and reasoning, inspiring widespread efforts to develop autonomous agents for real-world tasks (Wang et al., 2024). Real-world environments are inherently dynamic and change in response to the agent's actions, making the general knowledge in off-the-shelf LLMs often insufficient to ensure reliable agent behaviors (Xi et al., 2025a). To bridge this gap, a promising approach for agent optimization is to fine-tune LLMs on domain-specific data collected from agent–environment interactions, aligning their capabilities with task requirements.

Existing optimization methods primarily use task plans and execution trajectories to fine-tune LLMs and enhance agent capabilities in long-term planning and multi-step execution. For example, MPO (Xiong et al., 2025) decomposes complex goals into executable sub-steps as meta-plans, and WKM (Qiao et al., 2024) integrates world knowledge into the execution trajectory to optimize action selection. To improve the agent's task performance, LLM fine-tuning usu-

[1]College of Computer Science and Technology, National University of Defense Technology, Changsha, Hunan, China [2]State Key Laboratory of Complex & Critical Software Environment, Changsha, Hunan, China [3]National Key Laboratory of Parallel and Distributed Computing, Changsha, Hunan, China. Correspondence to: Dawei Feng <davyfeng.c@qq.com>.

*Proceedings of the 43rd International Conference on Machine Learning*, Seoul, South Korea. PMLR 306, 2026. Copyright 2026 by the author(s).

ally utilizes data with high task rewards, which are provided by the environment to measure task outcomes.

Despite notable advancements, these methods overlook the inherent incoordination between planning and execution in LLM-based agents. As illustrated in Figure 1, this incoordination is manifested primarily in two aspects: **1) Impractical Planning**, where plans seem logically sound but unexecutable in practice; and **2) Unfaithful Execution**, where executions deviate from the prescribed planned steps. These behavior patterns caused by incoordination are difficult for task rewards to capture and can propagate into the agent's optimization process. Fine-tuning LLMs directly on these incoordination data risks reinforcing flawed behavioral patterns (Shang et al., 2025; Sun et al., 2023; Xiong et al., 2024), as the model learns to mimic impractical plans and plan-deviated executions. In addition, our empirical results show that successful task trajectories exhibit stronger coordination compared to failed ones, highlighting the critical role of coordination in task completion. As a result, current optimization methods lead to suboptimal performance and limit further development of LLM-based agents.

To alleviate this issue, we propose **CoPE**, a novel framework that explicitly integrates planning-execution coordination into LLM-based agent optimization. CoPE consists of three stages: **1) Data Collection:** We integrate a self-refinement mechanism (Madaan et al., 2023) into the Monte Carlo Tree Search (MCTS) (Browne et al., 2012) to collect highly diverse plan-execution data. **2) Coordination Assessment:** We characterize this coordination with two measurable dimensions: *plan executability*, assessed via consistency across multiple execution trajectories, and *execution adherence*, quantified by the semantic similarity between the plan and its corresponding execution trajectory. **3) Agent Optimization:** We design an optimization method that explicitly leverages these coordination scores as sample weights during LLM fine-tuning, encourages the agent to learn better planning and execution policies. Through iterative optimization, CoPE strengthens the planning–execution coordination and empowers LLM-based agents to generate practical plans and execute them faithfully.

We conduct experiments on two challenging long-horizon, multi-turn interactive tasks, AlfWorld (Shridhar et al., 2021) and ScienceWorld (Wang et al., 2022). Extensive experiments demonstrate that CoPE achieves state-of-the-art performance, surpassing several strong baselines with an average reward of 93.65 in AlfWorld and 80.84 in ScienceWorld. Further analysis confirms that CoPE effectively enhances LLM agents coordination, increasing plan executability by 24.2% and execution adherence by 65.6% compared to vanilla agents. Notably, by leveraging improved coordination, CoPE agents complete tasks with shorter trajectories, reflecting superior execution efficiency.

Our main contributions are summarized as follows:

- We highlight the significance of coordinating between planning and execution for LLM-based agents, and empirically demonstrate its impact on task completion.
- We propose CoPE, a novel LLM-based agent optimization framework that assesses and incorporates planning-execution coordination into agent optimization.
- Extensive experiments demonstrate the effectiveness of CoPE, which outperforms the state-of-the-art in terms of task performance and efficiency.

## 2. Related Work

**Planning in LLM-based Agents.** Recent studies have explored the integration of planning with LLMs to enhance the long-horizon task-solving capabilities of intelligent agents. Research in (Yao et al., 2023; Shinn et al., 2023) primarily investigates implicit planning, where planning occurs through interleaved reasoning and action generation, without adequate consideration of the global task. In contrast, explicit planning provides high-level strategies by decomposing tasks into discrete sub-goals, aligning them to executable granularity, and executing them sequentially (Qiao et al., 2025; Huang et al., 2024). These studies improve planning by incorporating world knowledge (Qiao et al., 2024) and constrained action rules (Zhu et al., 2025), or by learning appropriate plans from expert demonstrations (Xiong et al., 2025). Other approaches (Lu et al., 2025; Erdogan et al., 2025) attempt to provide real-time guidance by prompting LLMs to dynamically update plans. However, these efforts predominantly focus on the planning phase while neglecting the adherence of subsequent execution, often leading to a discrepancy where well-designed plans fail to be effectively implemented in practice. In summary, existing work lacks explicit modeling the coordination between planning and execution of LLM-based agents, which hinders the successful task completion.

**Fine-tuning LLMs as Agent.** The standard practice of developing agents involves fine-tuning general LLMs on task-specific data (Zeng et al., 2024; Chen et al., 2024; Xi et al., 2025b). However, this paradigm relies on high-quality expert trajectories or manual annotation, which are costly and limited in quantity. To address the problem, recent studies (Zelikman et al., 2022; Song et al., 2024; Zohar et al., 2025) have adopted self-training techniques, where LLMs are trained on self-generated data. When combined with reject sampling fine-tuning (RFT) (Yuan et al., 2023), LLM agent capabilities are substantially enhanced, but their outputs often exhibit severe homogeneity (Kirk et al., 2024), which can trigger model collapse (Dohmatob et al., 2025) during iterative training process. Prior approaches mitigate this by enhancing output diversity via chain-of-thought prompting (Zelikman et al., 2022), MCTS (Zhang et al.,

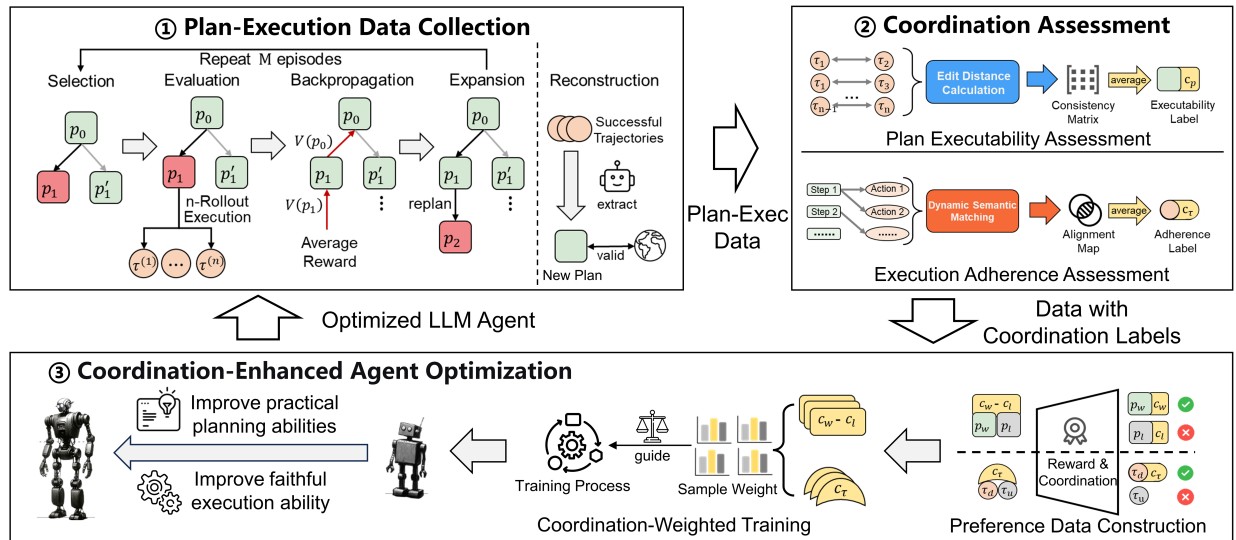

*Figure 2.* The overall framework of the proposed CoPE, which consists of three stages: Plan-Execution Data Collection, Coordination Assessment and Coordination-Enhanced Agent Optimization.

2024b), or other search algorithms to broaden the agent's exploration space. Besides, task rewards alone are often insufficient to accurately detect flawed behaviors (Zhai et al., 2025b), since even successful trajectories may contain redundant or failed intermediate steps (Shang et al., 2025; Sun et al., 2023; Zhu et al., 2025). In agent-based scenarios, such issues typically stem from impractical planning or unfaithful execution. Overall, our method introduces a self-refine planning mechanism within the MCTS to enhance the diversity of fine-tuning data, and mitigates the impact of inefficient behaviors on agent optimization through planning-execution coordination assessment.

## 3. Preliminaries

The agent task with environment feedback is formalized as a partially observable Markov decision process (POMDP) $(\mathcal{U}, \mathcal{S}, \mathcal{A}, \mathcal{O}, \mathcal{T}, r)$, where $\mathcal{U}$ represents the instruction space, $\mathcal{S}$ the state space, $\mathcal{A}$ the action space, $\mathcal{O}$ the observation space, $\mathcal{T} : \mathcal{S} \times \mathcal{A} \to \mathcal{S}$ the state transition function, and $r$ the reward function. Since the environment is partially observable, we can estimate the current state $s_t$ utilizing the concatenation of the task instruction $u$, the interactive history and other available information such as the task plan.

Given a task instruction $u \in \mathcal{U}$, the planning agent first generates a task plan $p \sim \pi_p(\cdot|u)$ with its policy $\pi_p$, which outlines a general strategy for task completion. The execution agent $\pi_e$ derives the initial action $a_0 \sim \pi_e(\cdot|u,p)$. The state transitions to $s_1 \in \mathcal{S}$, and the execution agent $\pi_e$ receives an observation $o_1 \in \mathcal{O}$. This process continues as the agent interacts with the environment until either the task is completed or the maximum number of steps is reached.

At each time step $t$, given the historical interactions $\tau_t =$ $(a_0, o_1, ..., a_{t-1}, o_t)$, the agent $\pi_e$ produces the next action $a_t \sim \pi_e(\cdot|u, p, \tau_t)$. The multi-step decision-making task can therefore be defined as:

$$\pi(\tau|u) = \pi_p(p|u) \cdot \prod_{t=0}^{T} \pi_e(a_t|u, p, \tau_t) \quad (1)$$

We denote $\tau$ as the whole plan-execution trajectory, $T$ as the total interaction steps. Note that the environment only provides the outcome reward $r(u, \tau) \in [0, 1]$.

The objective of LLM-based agents is to maximize rewards from the environment:

$$\max_{\pi} \mathbb{E}_{u \sim \mathcal{D}, \tau \sim \pi(\cdot|u)} \left[ r(u, \tau) \right], \quad (2)$$

where $\mathcal{D}$ represents the dataset containing task instructions.

## 4. Methodology

This section will begin by elaborating on the correlation between planning-execution coordination and the successful task completion of LLM-based agents, serving as the motivation for our approach. We then describe each stage of our proposed agent optimization framework, CoPE. The comprehensive workflow is illustrated in Figure 2.

### 4.1. Motivation

From the perspective of human intelligence, a practical and unambiguous plan mitigates execution uncertainty and promotes behavior consistency across multiple executions, and an execution that strictly adheres to the established plan more reliably ensures successful task completion. Building upon this intuition, we describe coordination from two aspects: *plan executability* and *execution adherence*. Figure 3

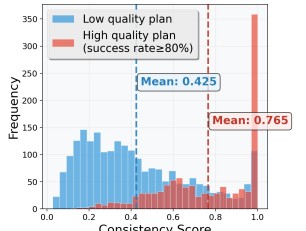 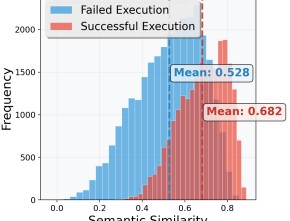

*(a)* Consistency across Multi-Execution Trajectories

*(b)* Semantic Similarity between Plan and Execution Trajectory

*Figure 3.* The distribution of planning-execution coordination.

presents the distribution of coordination scores in successful and failed trajectories in the Alfworld environment.

To assess plan executability, we employed an LLM-based agent to perform multiple task executions guided by the same plan, subsequently calculating the average consistency across these trajectories. As visually apparent in Figure 3a, high quality plans defined by a minimum success rate of 80% lead to more consistent executions. Their consistency distribution is heavily concentrated toward higher values, achieving a mean of 0.765, which significantly outperforms the 0.425 mean for low quality plans. Regarding execution adherence, we calculate a semantic similarity score between the trajectory and its corresponding plan. As shown in Figure 3b, successful trajectories exhibit a markedly stronger adherence to their plans. This is evidenced by a right-shifted distribution of semantic similarity and a higher mean value of 0.682, compared to 0.528 for failed trajectories.

The aforementioned analysis underscores that coordination between planning and execution in LLM-based agents is strongly associated with successful task completion. Our coordination assessment method is detailed in Section 4.3.

### 4.2. Self-Refining MCTS applied to Data Collection

To acquire high-quality and diverse fine-tuning data, we introduce Self-Refining MCTS, which integrates Monte Carlo Tree Search (MCTS) with the self-refinement capability of LLMs. During the tree search process, Self-Refining MCTS continuously refines task plans and conducts multi-round execution interactions, while also serving as the foundation for the subsequent coordination assessment.

We construct a search tree where each node represents the current plan, and each edge is defined as a replanning operation conducted based on the task trajectory of its parent node. Each node maintains a set of statistics:

$$V(p_t), N(p_t), R(p_t), \tag{3}$$

where $V(p_t)$ denotes the value function, representing the expected reward of node $p_t$ from replanning, $N(p_t)$ is the visit count of node $p_t$, and $R(p_t)$ refers to the average reward value obtained from multiple task executions under

the guidance of plan $p_t$. As shown in the first part of Fig. 2, our Self-Refining MCTS process performs $M$ episodes of four iterative phases: selection, evaluation, backpropagation and expansion, finally followed by reconstruction phase.

**Selection phase.** The purpose of the selection operation is to identify the most promising plan for the next replanning expansion. We use the Upper Confidence Bound applied to Trees (UCT) (Kocsis & Szepesvári, 2006) algorithm, which selects a node $p_t$ with the highest UCT value, ensuring a balance between exploration and exploitation:

$$UCT(p_t) = V(p_t) + \sqrt{\frac{\eta \ln N\big(parent(p_t)\big)}{N(p_t)}}, \tag{4}$$

where $\eta$ is the exploration weight, and $parent(p_t)$ denotes the parent node of $p_t$. Additionally, we employ dynamic pruning to avoid local optima: a node $p_t$ is considered fully explored and not selected if $R(p_t)$ reaches the maximum reward or any child node's $R(\cdot)$ exceeds $R(p_t)$.

**Evaluation phase.** Following the evaluation method in (Xiong et al., 2025; Zhang et al., 2024a), we evaluate the selected plan effectiveness by performing the $n$-Rollout Execution, which inserts a given plan into the execution agent's context and perform the task over $n$ independent trials. This results in $n$ task execution trajectories:

$$\{\tau^{(i)} \mid i = 1, \ldots, n\} \sim \pi_e(\tau \mid u, p) \tag{5}$$

For each execution trajectory $\tau^{(i)}$, an environmental reward $r(u, \tau^{(i)}) \in [0, 1]$ is returned. Thus, the quality of plan $p$ is determined by the agent's average reward, formulated as:

$$R(p) = \frac{1}{n} \sum_{i=1}^{n} r(u, \tau^{(i)}) \tag{6}$$

**Backpropagation phase.** Backpropagation updates the tree's statistics based on task rewards. For the node and recursively all their parent nodes, the visit count $N(p_t)$ is incremented by 1, and the value is propagated and updated from the node $p_t$ to the root node $p_0$ using the formula:

$$V(p_t) \leftarrow \frac{V(p_t)(N(p_t) - 1) + R(p_t)}{N(p_t)}. \tag{7}$$

This iterative update mechanism progressively refines the value function $V(\cdot)$ as an unbiased estimate of the replanning return, evaluating which nodes are likely to expand high-quality plans and guiding subsequent node selections.

**Expansion phase.** In the expansion phase, the selected node is expanded by generating $w$ new plans. The LLM-based agent takes the original plan $p_t$ and the trajectory $\tau$ with the lowest reward from $n$-Rollout Execution as input, performing a self-refine (Madaan et al., 2023) process to generate a new plan $p_{t+1}$:

$$p_{t+1} = \pi_p(u, p_t, \tau). \tag{8}$$

Replanning based on failed trajectories enables the agent to avoid previous mistakes and, compared to temperature-based sampling, enhances planning diversity for the same task (Zhai et al., 2025a). An example of replanning is provided in Appendix F. To control the growth of the search tree, we restrict the maximum depth $d$ of expansion.

**Reconstruction phase.** Due to the limitations of model capabilities, the agent often fails to generate effective plans for challenging tasks, which in turn reduces the scale of collected data and constrains the model's generalization ability after training. We argue that flawed plans can lead to occasional success by chance. Although such successful trajectories may be incoordinate, they encapsulate task-relevant insights that can be extracted by prompting LLMs to summary comprehensive plans. These resulting plans will then guide a new round of $n$-Rollout Execution, with their effectiveness validated through environmental interactions.

### 4.3. Coordination Assessment

Based on the motivation in Section 4.1, we conduct the co-ordination assessment of our collected plan-execution data. Specifically, for each plan, we measure its executability by the consistency of multiple executions; for each trajectory, we evaluate its adherence by calculating the semantic similarity between planned steps and executed actions.

**Plan Executability Assessment.** When a plan lacks comprehensiveness and is unrealistic, the execution trajectories tend to resort to random exploration, exhibiting high inconsistency. This scenario is referred to as **Impractical Planning** in Figure 1. In such cases, even successful task completion cannot be credited to the plan's contribution.

Based on execution trajectories $\{\tau^{(i)} \mid i = 1, \ldots, n\}$, we extract their agent's action sequences, and calculate pairwise distances with normalized Levenshtein distance (Yujian & Bo, 2007) to derive a consistency matrix. Each element in the matrix represents the consistency score between two trajectories, and we take the average of all elements as the continuous label $c_p \in [0, 1]$ to assess the plan executability.

**Execution adherence Assessment.** Execution trajectories that successfully complete the task may still deviate from the prescribed plan, as illustrated by the case of **Unfaithful Execution** in Figure 1. However, outcome-only rewards cannot effectively identify these data. To address this, we introduce an execution adherence assessment algorithm that performs dynamic semantic matching between planned steps and actual executed actions.

The detailed procedure is outlined in Algorithm 1. Given a plan consisting of steps $s_i$ and an execution trajectory composed of actions $a_j$, we first compute the cosine similarity of their word embedding vectors, denoted as $\mathrm{emb\_cos}(s_i, a_j)$. Then we define a rule-based matching $\mathrm{action\_match}(s_i, a_j)$,

---

**Algorithm 1** Step-Aligned Semantic Matching Algorithm.

**Input:** Plan steps $p = \{s_1, \ldots, s_n\}$, action sequence $\tau = \{a_1, \ldots, a_m\}$, weight coefficient $\alpha$, penalty constants $\lambda_1$ and $\lambda_2$

**for** $i = 1$ to $n$ **do**
    **for** $j = 1$ to $m$ **do**
        $M[i][j] \leftarrow (1 - \alpha) \cdot \mathrm{emb\_cos}(s_i, a_j) + \alpha \cdot \mathrm{action\_match}(s_i, a_j)$
    **end for**
**end for**
Initialize matrix $DP \in \mathbb{R}^{n \times m}$ with 0
**for** $i = 1$ to $n$ **do**
    **for** $j = 1$ to $m$ **do**
        $dp_{\text{match}} \leftarrow DP[i-1][j-1] + M[i][j]$
        $dp_{\text{skip-action}} \leftarrow DP[i][j-1] - \lambda_1$
        $dp_{\text{skip-step}} \leftarrow DP[i-1][j] - \lambda_2$
        $DP[i][j] \leftarrow \max(dp_{\text{match}}, dp_{\text{skip-action}}, dp_{\text{skip-step}})$
    **end for**
**end for**
**Output:** Alignment score $DP[n][m]/n$

---

which is 1 if $a_j$ belongs to a set of expected actions for $s_i$, and 0 otherwise. The expected action set is constructed by: (i) the exact keyword matching with the environment's action vocabulary (e.g. "goto", "take"), or (ii) if no keywords match, selecting the top $k$ actions ranked by cosine similarity (emb_cos) between $s_i$ and the textual description of each action. Details of rule-based matching are provided in Appendix B. The mixed similarity matrix $M$ is computed as a weighted combination of semantic and keyword matching scores, balanced by a fixed weight $\alpha \in [0, 1]$.

Subsequently, we employ a dynamic programming (Bellman, 1966) approach to find the optimal alignment map between each action and planned step. Let $DP[i][j]$ denote the optimal cumulative alignment score achievable by aligning the first $i$ steps of the plan with the first $j$ actions of the execution trajectory. Since planning and execution are both sequential, we introduce constants $\lambda_1$ and $\lambda_2$ to penalize the alignment of skipping an execution action and a planned step, respectively. Finally, we define the adherence score to the plan as $DP[n][m]/n$, and use this value as the continuous label $c_\tau \in [0, 1]$ for the execution trajectory $\tau$.

### 4.4. Coordination-Enhanced Agent Optimization

Recent study (Zhang et al., 2025) has shown that learning from preference samples can significantly enhance the capability of LLM-based agents. Based on task rewards and coordination scores, we construct preference pairs separately for planning and execution data, with coordination labels further serving as sample-level weights in agent optimization. The core principle of our agent optimization is to assign greater weight to plans that exhibit higher executability and execution trajectories that demonstrate stronger adherence.

**Preference Data Construction.** The task instruction $u$ serves as the input for preference pairs in Direct Preference Optimization (DPO) (Rafailov et al., 2023). For the plan preference pairs $(p_w, p_l)$ in each task, where $p_w$ is the positive example and $p_l$ is the negative example, are constructed based on two criteria: (i) $R(p_w) \geq \beta_p$ and $R(p_w) - R(p_l) \geq \beta_p/2$, where $\beta_p > 0$ is a reward threshold; (ii) among all plans $p$ with $R(p) \geq \beta_p$, the pair maximizes the coordination score difference. The coordination label for these planning pair is $c_w - c_l$, reflecting their relative thoroughness in task consideration.

Unlike plans, agent execution trajectories are multi-turn interactions. Therefore, we adopt the Kahneman–Tversky Optimization (KTO) (Ethayarajh et al., 2024), which eliminates the requirement for paired samples by building preference data directly from the desirable execution trajectory $\tau_d$ and the undesirable $\tau_u$. Similarly, for each plan, we select the trajectories satisfying (i) $r(u, \tau_d) \geq \beta_\tau$ and $r(u, \tau_d) - r(u, \tau_u) \geq \beta_\tau/2$, where $\beta_\tau > 0$ is a reward threshold; (ii) the pair maximizes the coordination score difference among all trajectories $\tau$ with $r(u, \tau) \geq \beta_\tau$. Additionally, to enable the agent to learn effective actions, we apply ***Execution Action Pruning*** to shorten the training trajectories by removing shared prefix actions from preference pairs. Specifically, for time step $h$, $\tau_d$ and $\tau_u$ have identical action sequences and share the same historical context $\tau_{\leq h}$, which will not be included in the loss calculation. Meanwhile, each task contains multiple trajectory pairs, enabling the agent to learn more comprehensive action preferences.

**Coordination-Weighted Training.** To mitigate the negative impact of incoordination data, we integrate coordination labels into the agent optimization process.

For the optimization of planning agent, we have the Coordination Weighted DPO (CW-DPO):

$$\mathcal{L}_{\text{CW-DPO}}(\pi_p, \pi_{\text{ref}}) = -\mathbb{E}_{(u, p_w, p_l) \sim \mathcal{D}}[(\frac{c_w - c_l + 1}{2})$$
$$\cdot \log \sigma(\beta \log \frac{\pi_p(p_w|u)}{\pi_{\text{ref}}(p_w|u)} - \beta \log \frac{\pi_p(p_l|u)}{\pi_{\text{ref}}(p_l|u)})] \quad (9)$$

where $c_w$ and $c_l$ refer to the plan executability labels $c_p$ of the respective plans and the $\frac{c_w - c_l + 1}{2} \in [0, 1]$, showing that preference pairs with larger differences exert a stronger influence on agent optimization. The reference policy $\pi_{\text{ref}}$ anchors the learned policy $\pi_p$, preventing its outputs from drifting far from the original behavioral distribution.

For execution preferences data, we concatenate $u$, $p$, and $\tau_{\leq h}$ to form the input context $x$, and take the segments of $\tau_w$ and $\tau_l$ after time step $h$ as the target outputs $y = \tau_{>h}$. With these definitions, the KTO loss is:

$$\mathcal{L}_{\text{KTO}}(\pi_e, \pi_{\text{ref}}) = \mathbb{E}_{(x,y) \sim \mathcal{D}}[\lambda_y - v(x, y)] \quad (10)$$

where

$$r_e(x, y) = \log \frac{\pi_e(y|x)}{\pi_{\text{ref}}(y|x)}, z_0 = \text{KL}\left(\pi_e(y'|x) \| \pi_{\text{ref}}(y'|x)\right)$$

$$v(x, y) = \begin{cases} \lambda_D \sigma\left(\beta(r_\theta(x, y) - z_0)\right) & \text{if } y \sim y_{\text{desirable}} \mid x \\ \lambda_U \sigma\left(\beta(z_0 - r_\theta(x, y))\right) & \text{if } y \sim y_{\text{undesirable}} \mid x \end{cases}$$

and $\lambda_y$ denotes $\lambda_D$ if $y$ is desirable positive example and $\lambda_U$ if $y$ is the undesirable negative example. The KL divergence also serves as a constraint. Additionally, we simultaneously perform supervised fine-tuning (SFT) on the desirable execution trajectories:

$$\mathcal{L}_{\text{SFT}}(\pi_e) = -\mathbb{E}_{(x,y) \sim \mathcal{D}}[c_\tau \cdot \log \pi_e(y \mid x)], y \sim y_{\text{desirable}} \mid x \quad (11)$$

where $c_\tau$ is the coordination label of $\tau_d$. Combining these objective, we optimize the execution agent with the Coordination Weighted KTO (CW-KTO):

$$\mathcal{L}_{\text{CW-KTO}}(\pi_e, \pi_{\text{ref}}) = \mathcal{L}_{\text{SFT}}(\pi_e) + \mathcal{L}_{\text{KTO}}(\pi_e, \pi_{\text{ref}}) \quad (12)$$

## 5. Experiment

### 5.1. Experimental Settings

**Tasks and Metrics.** Our experiments encompass two long-horizon, multi-step agent-based environments: *AlfWorld* (Shridhar et al., 2021) for household tasks and *ScienceWorld* (Wang et al., 2022) for scientific reasoning. In addition to seen tasks, both environments include unseen tasks to evaluate the agent's generalization ability. We evaluate agent performance with **Average Reward**, defined as the mean score across all test instances, and execution efficiency with **Trajectory Length**, measured by the average number of interaction steps.

**Baselines.** We employ GPT-4o (Hurst et al., 2024), Deepseek-Chat (DeepSeek-AI, 2025) and Gemini 2.5 (Comanici et al., 2025) as the Close-Sourced Models competitors. In addition, we perform comprehensive evaluations against several strong baselines. **ReAct** (Yao et al., 2023) serves as a foundational interactive style, representing the baseline performance prior to any optimization. The self-training method **RFT** (Yuan et al., 2023) applies SFT to self-generated trajectories filtered by task rewards. Other trial-and-error learning approaches such as **ETO** (Song et al., 2024) and the reinforcement learning framework **AGEN-TEVOL** (Xi et al., 2025b), also rely solely on task rewards for trajectory selection. For explicit planning optimization, we include meta-plan generation **MPO** (Xiong et al., 2025) and dynamic planning **PiplotRL** (Lu et al., 2025). In the realm of knowledge-enhanced agents, we include **KnowAgent** (Zhu et al., 2025) and **WKM** (Qiao et al., 2024). All baseline methods are optimized using an identical instruction task set $\mathcal{U}$.

*Table 1.* Performance Comparison. *"w/ Plan."* indicates whether the inference paradigm provides explicit guidance. The best and second best results for each model are in **bold** and underlined. Red represents the changes of CoPE relative to the optimal results in the baselines.

| Backbone Model | Method | w/ Plan. | AlfWorld | | | | ScienceWorld | | | |
|---|---|---|---|---|---|---|---|---|---|---|
| | | | Average Reward ↑ | | Trajectory Length ↓ | | Average Reward ↑ | | Trajectory Length ↓ | |
| | | | Seen | Unseen | Seen | Unseen | Seen | Unseen | Seen | Unseen |
| *Close-Sourced Models* | | | | | | | | | | |
| GPT-4o | ReAct | ✘ | 63.77 | 65.15 | 16.64 | 16.49 | 62.50 | 62.42 | 21.45 | 20.60 |
| Deepseek-Chat | ReAct | ✘ | 76.64 | 80.30 | 15.36 | 15.33 | 61.90 | 63.21 | 20.94 | 22.13 |
| Gemini 2.5-Pro | ReAct | ✘ | 66.91 | 66.92 | 17.41 | 18.26 | 78.36 | 78.33 | 20.85 | 21.83 |
| *Open-Sourced Models + Agent Optimization Methods* | | | | | | | | | | |
| Qwen2.5-7B | ReAct | ✘ | 52.99 | 58.06 | 15.70 | 15.61 | 35.63 | 33.31 | 21.94 | 22.18 |
| | RFT | ✘ | 69.00 | 72.24 | 16.40 | 15.75 | 40.73 | 39.88 | 21.74 | 21.35 |
| | ETO | ✘ | 68.96 | 72.40 | 16.29 | 16.61 | 43.15 | 45.41 | 20.46 | 21.34 |
| | MPO | ✔ | 79.29 | 79.85 | 13.96 | 14.18 | 57.71 | 58.18 | 22.33 | 24.29 |
| | KnowAgent | ✘ | 73.43 | 68.66 | 16.13 | 16.90 | - | - | - | - |
| | WKM | ✘ | 75.67 | 74.40 | 17.74 | 16.80 | 58.22 | 55.74 | 23.16 | 23.59 |
| | PiplotRL | ✔ | 70.40 | 71.20 | 17.34 | 17.44 | - | - | - | - |
| | AGENTEVOL | ✘ | 80.62 | 81.64 | 14.21 | 14.26 | 38.70 | 40.12 | 20.93 | 23.64 |
| | **CoPE (Our)** | ✔ | **88.00** +7.38 | **86.27** +4.63 | **11.26** -2.70 | **11.34** -2.84 | **64.67** +6.45 | **65.76** +7.58 | **20.18** -0.28 | **19.62** -1.72 |
| Qwen3-32B | ReAct | ✘ | 75.86 | 73.88 | 15.92 | 16.65 | 65.80 | 59.01 | 18.09 | 17.06 |
| | RFT | ✘ | 82.24 | 83.60 | 16.41 | 16.94 | 70.34 | 68.88 | 19.84 | 18.62 |
| | ETO | ✘ | 83.20 | 81.14 | 16.65 | 17.86 | 70.54 | 69.10 | 18.88 | 18.54 |
| | MPO | ✔ | 86.52 | 84.20 | 11.96 | 11.88 | 73.62 | 74.77 | 18.01 | 17.24 |
| | KnowAgent | ✘ | 76.94 | 76.44 | 17.93 | 17.35 | - | - | - | - |
| | WKM | ✘ | 77.24 | 75.27 | 17.32 | 16.81 | 66.45 | 58.12 | 23.40 | 22.74 |
| | PiplotRL | ✔ | 79.53 | 79.42 | 17.42 | 17.12 | - | - | - | - |
| | AGENTEVOL | ✘ | 85.74 | 85.36 | 12.27 | 14.55 | 74.64 | 70.44 | 17.91 | 17.30 |
| | **CoPE (Our)** | ✔ | **93.65** +7.13 | **92.27** +6.91 | **11.20** -0.76 | **11.60** -0.28 | **80.84** +6.20 | **81.30** +6.53 | **17.21** -0.70 | **16.58** -0.48 |

**Implementation Details** In our experiments, we used Qwen2.5-7B (Yang et al., 2024) and Qwen3-32B (Yang et al., 2025) as our backbone models. We first performed SFT on LLM to equip it with basic agent capabilities, and then proceeded with multiple rounds of iterative optimization using our optimization method. All optimizations of LLM-based agents were fine-tuned using LoRA (Hu et al., 2022) technology, with the low-rank dimension set to 64. Notably, KnowAgent and PiplotRL baselines were not implemented in ScienceWorld. All hyperparameters and detailed experimental settings are provided in Appendix C.

## 5.2. Main Results

We report the average scores from five independent runs for all evaluation metrics in Table 1.

**Overall Performance.** Our proposed CoPE achieves superior performance and consistently outperforms existing baselines across all datasets. On average, CoPE significantly exceeds the strongest baseline in AlfWorld by 6.01 with Qwen2.5-7B and 7.02 with Qwen3-32B in average reward, and gets similar performance gains of 7.02 and 6.37 in ScienceWorld, respectively. Compared to advanced closed-source LLM-based agents, CoPE-optimized models obtain better performance on specific tasks: Qwen2.5-7B scores 88.00 on the AlfWorld Seen, outperforming Deepseek-Chat's 76.64, and Qwen3-32B attains 81.30 on the ScienceWorld Seen, surpassing Gemini 2.5-Pro's 78.36.

**The effectiveness of explicit planning depends on both high-quality plans and tight integration with execution.** Although PiLotRL incorporates explicit planning, it offers

no clear advantage over ETO or RFT. In contrast, MPO enhances task performance by improving plan executability through meta-plan abstraction, and CoPE further advances this paradigm by strengthening adherence to planned steps, achieving higher performance via effective execution.

**Optimization methods relying solely on task rewards struggle to learn action preferences in long-horizon scenarios.** RFT, ETO, and AGENTEVOL exhibit only marginal performance gains under limited model capacity (e.g., Qwen2.5-7B in ScienceWorld), as their collected trajectories contain excessive inefficient or failed actions. In contrast, WKM mitigates this issue by injecting external expert knowledge, while CoPE enhances learning efficiency by introducing coordination-aware weights to explicitly reinforce planning-execution alignment.

**Execution Efficiency.** CoPE demonstrates superior execution efficiency over existing approaches, as evidenced by its achievement of the shortest trajectory length. Besides, we observe that performance gains might come at the cost of increasing trajectory lengths. This is attributed to the fact that more complex tasks usually require more action steps.

## 5.3. In-Depth Analysis

In this subsection, we conduct experiments with Qwen2.5-7B to study the impact of CoPE on agent behaviors, training data quality, iterative optimization process and a case study.

**CoPE significantly enhances the coordination between planning and execution in LLM-based agents.** To evalute agent behavior, we use the coordination assessment method

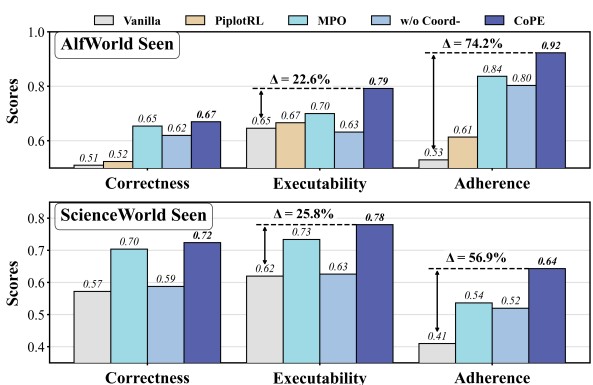

*Figure 4.* Multi-dimensional agent behavior scores.

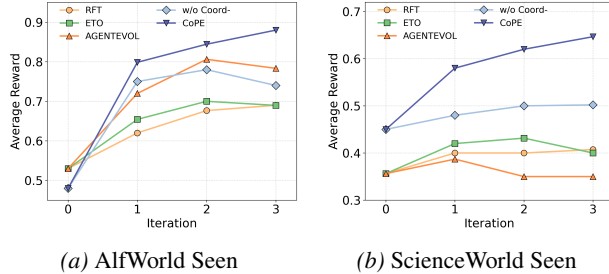

*(a)* AlfWorld Seen          *(b)* ScienceWorld Seen

*Figure 5.* Performance improvement through multiple iterations.

proposed in Section 4.3 to score the *plan executability* and *execution adherence*. Besides, we also employ the LLM-as-a-Judge approach (Gu et al., 2024) to assess the *Correctness*, which refers to the extent to which the plan fulfills the task requirement. For comparison, we introduce "*w/o Coord-*", a variant using only the original preference optimization without coordination modeling, and "*Vanilla*", which denotes the base LLM with ReAct prompting. As shown in Figure 4, compared to "*Vanilla*", CoPE achieves an average improvement of 24.2% in executability and 65.6% in adherence across two task scenarios. Moreover, the limited improvement of "*w/o Coord-*" underscores the importance of explicitly modeling coordination. Notably, while the MPO agent obtains competitive scores with CoPE in plan correctness and executability, its lower execution adherence translates to inferior overall task performance.

**Replanning from failed trajectories enhances the diversity of both plans and execution trajectories.** To investigate the impact of CoPE on training data, we analyze the quality and diversity of collected data. Quality is measured by the retention rate after filtering trajectories with outcome rewards greater than $\beta_\tau$; diversity is quantified via normalized Levenshtein distance (NLD) and cosine similarity between multiple trajectories, reporting the maximum inter-trajectory divergence for each task. Additionally, we introduce a variant baseline, "*w/o Replanning*" in which node expansion in MCTS relies solely on temperature-based sampling. The statistics are shown in Table 2. CoPE-collected

*Table 2.* Statistics of the collected training data.

| | Retention Rate ↑ | NLD ↑ | Cosine Similarity ↓ |
|---|---|---|---|
| **AlfWorld** | | | |
| CoPE | **57.43%** | **0.79** | **0.71** |
| *w/o Replanning* | 53.69% | 0.56 | 0.80 |
| RFT | 33.90% | 0.32 | 0.88 |
| ETO | 35.67% | 0.45 | 0.75 |
| AGENTEVOL | 47.44% | 0.48 | 0.76 |
| **ScienceWorld** | | | |
| CoPE | **28.63%** | **0.86** | **0.63** |
| *w/o Replanning* | 20.86% | 0.44 | 0.84 |
| RFT | 15.46% | 0.47 | 0.79 |
| ETO | 16.93% | 0.45 | 0.84 |
| AGENTEVOL | 12.45% | 0.50 | 0.83 |

data exhibit higher quality and diversity, as evidenced by a higher retention rate, greater NLD, and lower semantic similarity. The replanning mechanism in CoPE substantially increases planning diversity, nearly halving the Self-BLEU (Zhu et al., 2018) score from 0.48 to 0.25 (lower Self-BLEU indicates higher diversity), thereby enhancing the diversity of execution trajectories under plan guidance. Furthermore, replanning enables agents to avoid previous errors, leading to improved trajectory quality, such as an increase in AlfWorld retention rate from 53.69% to 57.43%.

**Directly fine-tuning LLMs on incoordination data undermines performance gains in multi-round iterations.** We analyze performance improvement of our approach on Qwen-2.5-7B relative to these baselines across multiple optimization iterations, which is presented in Figure 5. In both scenarios, the "*w/o Coord-*" variant shows initial improvement in early iterations; however, the agent's flawed behavior degrades the quality of collected data, leading to rapid performance saturation or even degradation in subsequent iterations. In contrast, CoPE shows consistent performance improvement over three rounds.

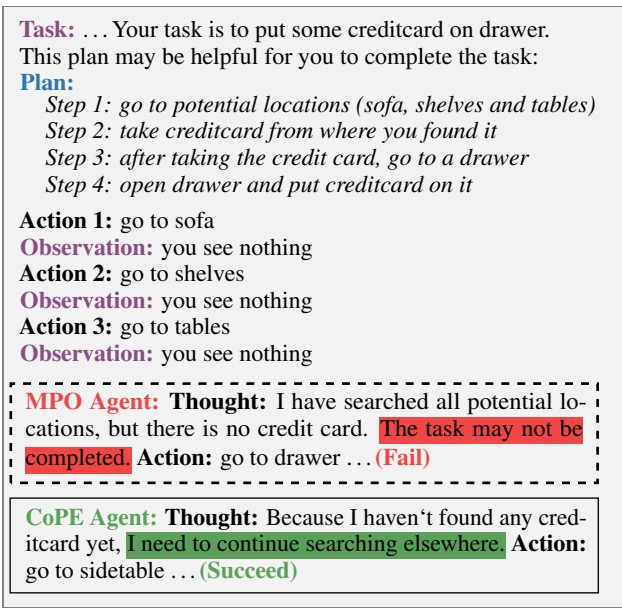

*Figure 6.* Case study on the AlfWorld task.

**Case Study.** Figure 6 illustrates a case study in the AlfWorld environment, where the goal is to place some credit cards on a drawer. The provided high-level plan suggests searching potential locations, such as the sofa, shelves, tables. LLM agents initially visits these locations but observes nothing in each step, and their subsequent decision-making differs markedly. The MPO agent abandons the search for task item prematurely, deviating from the original plan, leading to failure. In comparison, the CoPE agent adheres to the task plan throughout the search process and ultimately completes the task successfully.

## 6. Conclusions

This paper presents CoPE, a novel framework that explicitly integrates coordination between planning and execution into the LLM-based agent optimization process. CoPE mitigates the suboptimal agent performance and erroneous behavior patterns caused by impractical plans and plan-deviated trajectories in fine-tuning data through coordination assessment and coordination-weighted training. Extensive experiments demonstrate that CoPE outperforms existing methods in terms of task performance and agent efficiency, particularly in maintaining instruction adherence over long-horizon tasks, highlighting the importance of enhancing planning-execution coordination in LLM-based agents.

## Acknowledgments

This work was partially supported by National Science and Technology Major Project (2023ZD0121101), the Open Fund of National Key Laboratory of Parallel and Distributed Computing (PDL) (NO.2024-KJWPDL-02) and the Science and Technology Innovation Program of Hunan Province (No.2023RC1005).

## Impact Statement

This work exposes the crucial coordination between planning and execution in LLM-based agents when handling long-range multi-step tasks, and demonstrates how to enhance this coordination through LLM fine-tuning methods. These findings emphasize the urgent need for intelligent agents to comprehensively plan and faithfully execute instructions in long-term workflow tasks such as embodied manipulation and scientific experiment.

Our research aims to advance the effectiveness and robustness of LLM-based agent systems in real-world scenarios. Future efforts should focus on incorporating real-time feedback into adaptive replanning and other fine-grained data assessment dimensions.

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

# Appendix

The appendix of this paper is organized as follows.

- Appendix A provides an ablation study, hyperparameter analysis, and other supplementary experiments.
- Appendix B discusses the limitations of CoPE in other scenarios and presents the supplementary CoPE's coordination assessment.
- Appendix C presents detailed experimental settings, including dataset descriptions, baseline configurations and hyperparameters.
- Appendix D analyzes the agent optimization algorithms in CoPE.
- Appendix E analyzes the time complexity and practical efficiency of CoPE.
- Appendix F presents a replanning case and an example of calculating the execution adherence score using Algorithm 1.
- Appendix G provides the complete prompts used by CoPE.

## A. Additional Experiment Results

### A.1. Ablation Study

To evaluate the effectiveness of our agent optimization method, we performed ablation studies based on Qwen2.5-7B to isolate the individual contributions of key components in both the planning and execution agents. The influence of each eliminated component is detailed below.

*Table 3.* Results of Ablation Study with Qwen2.5-7B.

| Module | AlfWorld Seen | ScienceWorld Seen |
|---|---|---|
| Full* | 88.00 | 64.67 |
| **Planning Agent** | | |
|   w/o Weighting | 84.47 | 61.22 |
|   s RBW | 82.60 | 58.14 |
|   s IPO Loss | 80.93 | 58.28 |
| **Execution Agent** | | |
|   w/o SFT Loss | 81.81 | 51.63 |
|   w/o EAP | 76.65 | 55.20 |
|   s constant weight | 72.49 | 45.11 |

\* "Full" denotes the complete version of CoPE.
"s" denotes the substitution made to the agent optimization.

For planning agent optimization, the configurations are as follows:

- w/o Weighting: Removes the weight term $\frac{c_w - c_l + 1}{2}$ in Equation 9, which means using the original DPO algorithm for preference optimization.
- s Reward-Based Weighting (RBW): Substitutes the coordination label $c_w$ and $c_l$ with the average outcome rewards obtained from environmental feedback. This indicates that samples with higher task rewards will receive more attention.
- s IPO Loss: Substitutes the loss function with an IPO-style (Azar et al., 2024) formulation which integrates the coordination label as a difference term into the DPO objective.

For execution agent optimization, the configurations are as follows:

- w/o SFT Loss: Removes the loss term $\mathcal{L}_{sft}$ in Equation 12, which means using the KTO algorithm for preference optimization.
- w/o Execution Action Pruning (EAP): Constructs trajectory-level preference pairs directly from execution data, skipping the removal of similar prefix actions.
- s constant weight: Substitutes the SFT weight term $c_\tau$ in Equation 11 with a constant value in $[0.01, 0.1, 0.5, 1.0]$ and reports their highest performance.

We can have the following observations from the ablation results in Table 3. First, each component in our approach contributes to the overall performance, since eliminating any of them would result in performance degradation. Second,

the practice of weighting samples by reward values proves ineffective for planning agents in selecting high-value data, evidenced by the outperformance of the "w/o Weighting" over the "s RBW" method. Moreover, in our experimental setting, methods that incorporate coordination scores as implicit rewards, such as IPO–style loss, fail to effectively exploit this coordination information, resulting in reduced agent performance. Third, optimizing an execution agent by blindly applying a constant weight to the SFT loss can lead to downgrading the method to an RFT-like approach. Additionally, in multi-turn interactions, the redundant prefixes in trajectory-level sample pairs hinder the learning of fine-grained action preferences, underscoring the necessity of EAP.

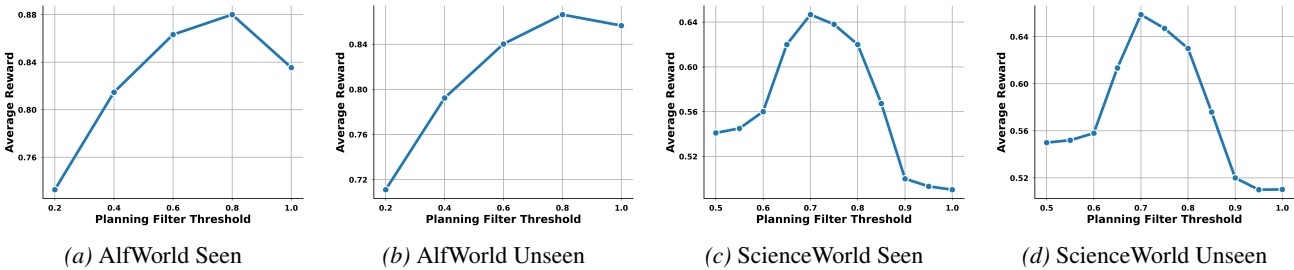

*(a)* AlfWorld Seen     *(b)* AlfWorld Unseen     *(c)* ScienceWorld Seen     *(d)* ScienceWorld Unseen

*Figure 7.* The effect of filter thresholds $\beta_p$ under Average Reward metrics.

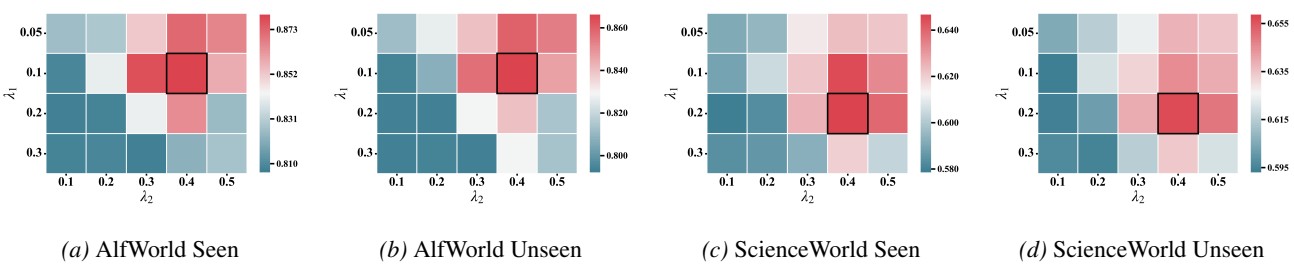

*(a)* AlfWorld Seen     *(b)* AlfWorld Unseen     *(c)* ScienceWorld Seen     *(d)* ScienceWorld Unseen

*Figure 8.* The effect of penalty constants $\lambda_1$ and $\lambda_2$ under Average Reward metrics.

### A.2. Hyperparameter Analysis

In CoPE, we introduce three critical hyperparameters, namely the trajectory filter threshold $\beta_p$ in agent optimization, and the penalty constants $\lambda_1$ and $\lambda_2$ in execution adherence assessment. Their value sensitivities are evaluated to facilitate future application of our proposed framework.

*1) Effect of filter threshold $\beta_p$:* We first investigate the impact of the hyperparameter $\beta_p$ in the data filtering process, which governs the quality and quantity of data used for agent optimization. Figure 7 illustrates the performance of CoPE in terms of average reward. We observe that, in the AlfWorld environment, agent performance improves as the threshold increases, peaking at $\beta_p = 0.8$ before declining; the optimal threshold in the ScienceWorld environment is $\beta_p = 0.7$. The results further indicate that raising the threshold enhances the quality of planning data, but an excessively high threshold drastically reduces the amount of training data, resulting in performance degradation.

*2) Effect of penalty constants $\lambda_1$ and $\lambda_2$:* To investigate under what conditions our proposed execution evaluation algorithm can effectively model the execution adherence, we vary the hyperparameters $\lambda_1$ and $\lambda_2$ over the ranges $\{0.05, 0.1, 0.2, 0.3\}$ and $\{0.1, 0.2, 0.3, 0.4, 0.5\}$, respectively, and report performance across all datasets in Figure 8 with the optimal cases highlighted by a black box.

From Figure 7 and Figure 8, we make the following observations: First, the optimal values of $\lambda_1$ and $\lambda_2$ differ across environments—for instance, AlfWorld achieves peak performance at $(\lambda_1, \lambda_2) = (0.1, 0.4)$, while ScienceWorld performs best at $(0.2, 0.4)$. Second, the ratio $\lambda_2/\lambda_1$ should be set greater than 1, indicating a stronger intolerance of skipping planned steps during semantic matching.

### A.3. Analysis of Invalid Action

We compared against optimization methods that filter trajectories solely based on rewards, evaluating the likelihood of generating invalid actions. Results are shown in Table 4, in simpler environments like AlfWorld, methods such as RFT, ETO,

and AGENTEVOL collect higher-quality data via task rewards, yielding optimized agents that generate fewer ineffective actions. In contrast, in the more challenging ScienceWorld setting, task rewards poorly reflect the efficacy of intermediate steps, leading optimized agents to learn ineffective behaviors.

Table 4. Invalid action rates with Qwen2.5-7B. Here *invalid* action refers to the inability to correctly interact with the environment.

| Method | AlfWorld | ScienceWorld |
|---|---|---|
| CoPE | 0.37% | **2.67%** |
| RFT | 0.85% | 10.78% |
| ETO | 0.62% | 8.35% |
| AGENTEVOL | **0.33%** | 9.76% |

## A.4. Analysis of Generalizability

To demonstrate generalization beyond environments with fixed action spaces, we evaluated CoPE on 2WikiMultihopQA(Ho et al., 2020). This benchmark features a flexible action space where agents must iteratively search for evidence using LLM-generated queries. We sampled 1,000 samples as the training set and 100 samples as the testing set, and used answer exact match as the reward. As shown in Table 5, CoPE achieves superior performance in 2WikiMultihopQA with backbone model Qwen2.5-7B. Furthermore, by setting $\alpha = 0$ in Algorithm 1, we disabled the rule-based matching component to rely solely on embedding similarity. CoPE still outperformed baseline methods, showing that its effectiveness stems from coordination rather than dataset-specific heuristics, thus validating its generalizability to dynamic environments.

Table 5. Performance comparison in 2WikiMultihopQA.

| Model | Method | Average Reward |
|---|---|---|
| GPT-4o | ReAct | 57.31 |
| Deepseek-Chat | ReAct | 61.72 |
| Gemini2.5-Pro | ReAct | 51.60 |
| | ReAct | 40.37 |
| | ETO | 52.39 |
| Qwen2.5-7B | RFT | 49.76 |
| | AGENTEVOL | 54.35 |
| | CoPE (Our) | **67.74** |

## A.5. Analysis of Agent Optimization Effectiveness

To demonstrate the effectiveness of incorporating coordination into agent optimization, we trained baseline models on the same data generated during CoPE's final iteration on AlfWorld. However, this data is incompatible with the inference paradigms of baselines such as KnowAgent, WKM, and PiPlotRL. As shown in Table 6, while other methods exhibit marginal gains, CoPE maintains significantly superior performance.

Additionally, we evaluated five Qwen2.5-7B execution agents with different random seeds in AlfWorld Seen. The resulting average rewards were 88.00±0.56, 88.12±0.83, 87.45±0.32, 86.98±0.96, and 87.43±0.51, respectively. The negligible variance across these five independent runs demonstrates the training stability of our weighting scheme.

Table 6. Performance comparison with the same training data in AlfWorld.

| Model | Method | Seen | Unseen |
|---|---|---|---|
| | RFT | 71.26 | 72.79 |
| | ETO | 73.43 | 74.29 |
| Qwen2.5-7B | MPO | 80.46 | 81.34 |
| | AGENTEVOL | 82.74 | 82.06 |
| | CoPE (Our) | **88.00** | **86.27** |

# B. Additional Discussions

## B.1. Limitations

Our method demonstrates strong performance under multi-step, long-horizon tasks that typically require multi-factor planning and involve longer interaction trajectories, where the incoordination in LLM agents becomes particularly pronounced, as illustrated in Figure 3 and Figure 9 with backbone model Qwen3-32B. However, in short-term tasks, this incoordination between planning and execution does not necessarily correlate with task completion.

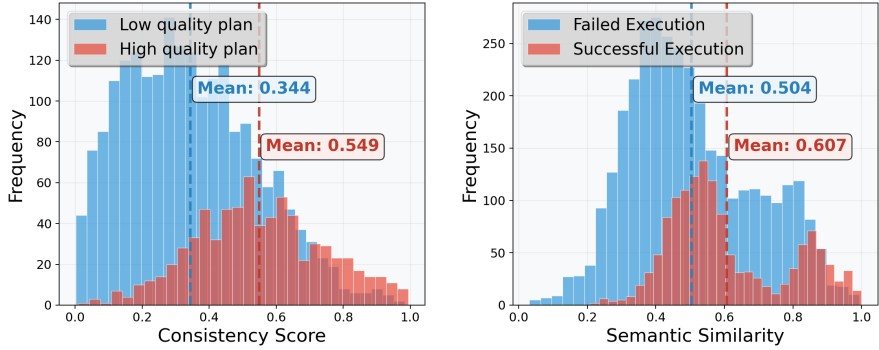

*(a)* Consistency across Multi-Execution Trajectories

*(b)* Semantic Similarity between Plan and Execution Trajectory

*Figure 9.* The distribution of planning-execution coordination in ScienceWorld.

Table 7 presents our analysis of the incoordination phenomenon in two additional agent tasks: WebShop (Yao et al., 2022) and HotPotQA (Yang et al., 2018). The column *Reasoning Steps* means the average trajectory length required by an oracle to complete the task (Wang et al., 2023; Cao et al., 2025; Lin et al., 2023). *Consistency Gap* and *Similarity Gap* quantify the differences between successful and failed trajectories in terms of plan executability and execution adherence, respectively, with the same setting as in Figure 3. It can be observed that for tasks with shorter reasoning steps, both consistency and similarity gap approach zero, suggesting incoordination is not the primary bottleneck for task completion in these scenarios.

Furthermore, we observe that explicit planning tends to be detrimental for short-horizon tasks. This is not a specific flaw of our approach but rather a general characteristic of planning in LLM agents. To validate this, we conducted additional experiments with Qwen2.5-7B, which performed worse when planning was enabled. Specifically, on HotPotQA, the reward decreased from 34.92 (without planning) to 28.25 (with planning), and on WebShop, it dropped from 44.20 to 39.66.

*Table 7.* The coordination gap between successful and failed trajectories under different agent scenarios.

| Dataset | Reasoning Steps | Consistency Gap | Similarity Gap |
|---|---|---|---|
| HotPotQA (Yang et al., 2018) | 2.68 | 0.01 | -0.05 |
| WebShop (Yao et al., 2022) | 3.64 | 0.02 | -0.03 |
| AlfWorld (Shridhar et al., 2021) | 7.97 | 0.34 | 0.15 |
| ScienceWorld (Wang et al., 2022) | 11.76 | 0.21 | 0.10 |

## B.2. Analysis of Sensitivity

We analyze the sensitivity of planning-execution coordination to the rollout number $n$ of each plan. As shown in Table 8, when the rollout number $n$ is 3, the discrepancy in coordination between successful and failed trajectories becomes sufficiently significant, providing reliable signals for subsequent agent optimization.

*Table 8.* Impact of rollout number $n$ on coordination discrepancy in AlfWorld.

| Rollout Number $n$ | 3 | 5 | 7 |
|---|---|---|---|
| Consistency Gap | 0.27 | 0.34 | 0.36 |
| Similarity Gap | 0.08 | 0.15 | 0.16 |

### B.3. Coordination Assessment

For a plan-execution trajectory $\tau = (u, p, a_0, o_1, ..., a_{t-1}, o_t)$, where $u$ is a task instruction, $p$ is a task plan, $a$ is the action taken by agents, $o$ is the observation of environmental feedback, and $t$ is the time step at which the task is completed or reaching the maximum steps. During assessment, we extract the sequence of valid actions sequences $(a_0, a_1, ...)$ from each trajectory.

**Plan executability.** To evaluate the generalizability of plan executability as an evaluation paradigm, we examined the ranking consistency of three metrics across multiple plans: normalized Levenshtein distance (Yujian & Bo, 2007), Self-BLEU (Zhu et al., 2018), and ROUGE-L (Lin, 2004). Specifically, we randomly sampled 100 AlfWorld tasks with 10 distinct plans per task, ranked plans by lower normalized Levenshtein distance and higher Self-BLEU/ROUGE-L, and measured ranking agreement with Spearman's correlation.

Formally, let $rank^{(1)} = (rank_1^{(1)}, \ldots, rank_{10}^{(1)})$ and $rank^{(2)} = (rank_1^{(2)}, \ldots, rank_{10}^{(2)})$ denote the rankings of the 10 plans under two metrics . The Spearman correlation (Spearman, 1987) between them is defined as:

$$\rho_{rank} = 1 - \frac{6 \sum_{i=1}^{10} \left( rank_i^{(1)} - rank_i^{(2)} \right)^2}{10 \cdot (10^2 - 1)}.$$

A high positive $\rho_{rank}$ (close to 1) indicates strong agreement in how the two metrics rank plan executability. As shown in Table 9, the high Spearman correlations among the three metrics indicate strong agreement in their assessment of plan executability, demonstrating that this notion is robust across diverse evaluation metrics.

*Table 9.* Spearman's rank correlation ($\rho_{rank}$) between different plan executability metrics in AlfWorld.

|  | **Levenshtein** | **ROUGE-L** | **Self-BLEU** |
|---|---|---|---|
| **Levenshtein** | 1.00 | 0.84 | 0.88 |
| **ROUGE-L** | - | 1.00 | 0.81 |
| **Self-BLEU** | - | - | 1.00 |

**Execution Adherence.** As illustrated in Table 15, we present the action vocabulary and its textual descriptions used for the rule-based matching in Section 4.3. In our experimental scenarios, the expected number of actions in *action_match* $k$ is set to 2. All text vector similarity computations, including emb_cos, are implemented by the all-MiniLM-L6-v2[1] model.

**LLM Judgement.** To validate the effectiveness of our method in capturing coordination between planning and execution, we examined their agreement with LLM-generated executability and adherence scores by measuring Spearman's rank correlation $\rho_{score}$. Following the same prompts in (Lu et al., 2025), we employ the LLM-as-a-Judge evaluation approach with multiple advanced agents for scoring.

*Table 10.* Spearman's score correlation ($\rho_{score}$) between LLMs and different methods in ScienceWorld.

| Method | GPT-4o | Deepseek-Chat | Gemini 2.5-Pro |
|---|---|---|---|
| **Plan** |  |  |  |
| Reward-based | 0.28 | 0.21 | 0.20 |
| Consistency-based (Our) | **0.72** | **0.70** | **0.73** |
| **Execution** |  |  |  |
| Vector-Similarity-based | 0.28 | 0.25 | 0.21 |
| Rule-Matched | 0.24 | 0.19 | 0.25 |
| Step-Aligned (Our) | **0.83** | **0.77** | **0.81** |

## C. Details of Experimental Setup

As summarized in Table 10, "Reward-based" means using average task-outcome rewards to measure plan executability; "Vector-Similarity-based" denotes adherence assessment relying solely on cosine similarity of vector embeddings, whereas "Rule-Matched" indicates assessment solely via rule-based matching of expected actions. It is observed that the coordination

---

[1] https://huggingface.co/sentence-transformers/all-MiniLM-L6-v2

*Table 11.* Statistics of the evaluation datasets.

| Dataset | #Train | #Test-Seen | #Test-Unseen | #Actions | #Objects |
|---|---|---|---|---|---|
| AlfWorld | 1,991 | 140 | 134 | 11 | 73 |
| ScienceWorld | 1,483 | 194 | 211 | 25 | 195 |

# denotes the number of this item.

scores computed by our method exhibit a significant correlation with the LLM ratings, demonstrating the effectiveness of our modeling approach in capturing the relationship between planning and execution. Compared to employing advanced LLMs for scoring, our method is low-cost and easier to implement.

## C.1. Datasets

*AlfWorld* (Shridhar et al., 2021) requires the agent to comprehend high-level user instructions and solve household tasks by navigating through rooms, manipulating objects, and executing correct action sequences. In addition to seen tasks, AlfWorld also includes unseen tasks to evaluate the agent's generalization ability. The reward of AlfWorld is binary 0 or 1, indicating whether the agent has completed the task or not.

*ScienceWorld* (Wang et al., 2022) is designed to simulate scientific reasoning scenarios, with the goal of testing an agent's ability to solve problems aligned with elementary school science curricula, covering areas such as object conductivity or basic chemical reactions. It also includes both seen and unseen parts and a dense reward function with values in $[0, 1]$ to evaluate the agent's completion of tasks.

Table 11 presents the statistics of the two datasets. The abundance of executable actions and interactive objects makes multi-step planning in these environments particularly challenging. In addition to an in-distribution "Test-Seen" set, both ALFWorld and ScienceWorld provide "Test-Unseen" sets that include out-of-distribution variations, which enables the evaluation of our framework's generalization capabilities.

## C.2. Baselines

- **ReAct** (Yao et al., 2023) is an interactive style that allows LLM to construct a complete series of actions to achieve expected goals. It is adopted by all baselines in our experiments, and we consider it as the basic performance of the agent before any optimization.

- **RFT** (Yuan et al., 2023) is the most classic self-training method that enhances agent's capabilities by applying supervised fine-tuning (SFT) to high-quality and self-generated trajectories filtered through environmental outcome rewards.

- **ETO** (Song et al., 2024) proposed a trial-and-error learning paradigm that pairs failure trajectories from the agent's exploration with expert demonstrations, forming trajectory-level preference data to optimize agent behavior.

- **MPO** (Xiong et al., 2025) represents an explicit planning approach that also employs a trial-and-error learning paradigm to refine the planner, enabling the formation of highly abstract meta-plans.

- **KnowAgent** (Zhu et al., 2025) constructs action knowledge by capturing constraint relationships between candidate actions and explicitly incorporates this structured knowledge into the reasoning process.

- **WKM** (Qiao et al., 2024) integrates world models into the agent reasoning process, leveraging task knowledge that captures scenario requirements and state knowledge that describes situational contexts to plan agent's behaviors.

- **PiplotRL** (Lu et al., 2025) introduce an adaptive global plan-based agent paradigm, which simultaneously optimizes both planning and execution agents through reinforcement learning to enhance end-to-end task performance.

- **AGENTEVOL** (Xi et al., 2025b) leverages the environmental outcome reward as the weight coefficient of trajectory data for policy optimization in reinforcement learning, iteratively improving the agent over multiple rounds.

*Table 12.* Hyperparameter Settings for AlfWorld and ScienceWorld

| Hyperparameter | AlfWorld | ScienceWorld |
|---|---|---|
| Rollout number $n$ | 5 | 5 |
| Tree width $w$ | 2 | 2 |
| Maximum tree depth $d$ | 4 | 4 |
| Weighting coefficient $\alpha$ | 0.7 | 0.7 |
| MCTS episodes number $M$ | 10 | 15 |
| Filtering thresholds $(\beta_p, \beta_\tau)$ | (0.8, 1.0) | (0.7, 0.8)) |
| Max plan pairs per task $p_{max}$ | 20 | 20 |
| Max execution pairs per task $e_{max}$ | 30 | 30 |
| Penalty constants $(\lambda_1, \lambda_2)$ | (0.1, 0.4) | (0.2, 0.4) |
| Sample Temperature | 1.0 | 1.0 |

### C.3. Hyperparameters

In our experiments, we use Qwen2.5-7B[2] and Qwen3-32B[3] as our backbone models. As shown in Table 12, our Self-Refining MCTS algorithm is configured with $n = 5$ rollout execution during the evaluation phase and a maximum tree depth $d = 4$ and the tree width $w = 2$ during the expansion phase. In coordination-aware data augmentation, the weighting coefficient $\alpha$ in similarity matrix $M$ is 0.7, prioritizing rule-based matching. In AlfWorld, the Self-Refining MCTS performs $M = 10$ episodes with data filtering thresholds $\beta_p = 0.8$ and $\beta_\tau = 1.0$ for planning and execution data respectively, while in ScienceWorld it conducts $M = 15$ episodes with corresponding thresholds of $\beta_p = 0.7$ and $\beta_\tau = 0.8$. These same threshold configurations for trajectories are consistently applied in other baseline method. To prevent overfitting, we select at most $p_{max}$ plan pairs for each plan and $e_{max}$ trajectory pairs for each task instruction, with half drawn based on reward and half on coordination score. Duplicate samples arising from overlapping selections are retained only once. During the reasoning process, the sampling temperature of all LLM agents was set to 1.0. The other default hyperparameters for all methods are set as suggested by the corresponding papers.

## D. Theoretical Analysis

### D.1. Optimization Objective for the Planning Policy

We begin by formulating the constrained reinforcement learning objective for the planning policy $\pi_p$. Following the trajectory decomposition of Equation 1 in Section 3, the joint distribution is factorized as:

$$\pi(\tau|u) = \pi_p(p|u) \cdot \pi_e(\tau|u, p) \tag{13}$$

The primary goal for $\pi_p$ is to maximize the expected environmental reward:

$$\max_{\pi_p} \mathbb{E}_{p \sim \pi_p(p|u)} \left[ \mathbb{E}_{\tau \sim \pi_e(\cdot|u,p)}[r(u, \tau)] \right] \tag{14}$$

Let $r(u, p)$ denote the latent reward for a plan $p$ and the instruction $u$, As specified in Equation 6, CoPE approximates this value via $n$-rollout Monte Carlo sampling:

$$r(u, p) \approx \mathbb{E}_{\tau \sim \pi_e(\cdot|u,p)}[r(u, \tau)] \tag{15}$$

To ensure training stability and prevent the policy from collapsing, we optimize $\pi_p$ under a Kullback-Leibler (KL) divergence

---

[2]https://huggingface.co/Qwen/Qwen2.5-7B-Instruct
[3]https://huggingface.co/Qwen/Qwen3-32B

constraint. The regularized objective is formulated as:

$$
\max_{\pi_p} \mathbb{E}_{p \sim \pi_p(p|u)}[r(u,p)] - \beta \mathrm{KL}\left(\pi_p(p|u) \| \pi_{\mathrm{ref}}(p|u)\right)
$$

$$
= \max_{\pi_p} \mathbb{E}_{p \sim \pi_p(p|u)}\left[r(u,p) - \beta \log \frac{\pi_p(p|u)}{\pi_{\mathrm{ref}}(p|u)}\right]
$$

$$
= \max_{\pi_p} \mathbb{E}_{p \sim \pi_p(p|u)}\left[\beta \log \exp\left(\frac{r(u,p)}{\beta}\right) - \beta \log \frac{\pi_p(p|u)}{\pi_{\mathrm{ref}}(p|u)}\right] \tag{16}
$$

$$
= \max_{\pi_p} \beta\, \mathbb{E}_{p \sim \pi_p(p|u)}\left[\log \frac{\pi_{\mathrm{ref}}(p|u)\exp\left(r(u,p)/\beta\right)}{\pi_p(p|u)}\right].
$$

By defining the partition function $Z(u) = \sum_p \pi_{\mathrm{ref}}(p|u)\exp\left(\frac{r(u,p)}{\beta}\right)$, the objective can be rewritten in terms of the Gibbs distribution $\pi^*(p|u) = \frac{1}{Z(u)}\pi_{\mathrm{ref}}(p|u)\exp\left(\frac{r(u,p)}{\beta}\right)$. Substituting this into the objective:

$$
\max_{\pi_p} \beta\mathbb{E}_{p \sim \pi_p}\left[\log \frac{Z(u)\pi^*(p|u)}{\pi_p(p|u)}\right] = \max_{\pi_p} \beta\left[\log Z(u) - \mathrm{KL}(\pi_p \| \pi^*)\right] \tag{17}
$$

Since $Z(u)$ does not depend on $\pi_p$, the objective is maximized when $\mathrm{KL}(\pi_p \| \pi^*)$ is minimized. By Gibbs' inequality, the optimal closed-form solution is given by:

$$
\pi^*(p|u) = \frac{1}{Z(u)}\pi_{\mathrm{ref}}(p|u)\exp\left(\frac{1}{\beta}r(u,p)\right) \tag{18}
$$

where $\beta > 0$ is a parameter controlling the strength of the KL penalty.

### D.2. Gradient Analysis of CW-DPO

To understand the optimization dynamics of CW-DPO, we examine the gradient of the objective function. We define the coordination weight as $\omega = \frac{c_w - c_l + 1}{2}$ and let $u_\theta$ be the log-ratio difference:

$$
u_\theta = \beta \log \frac{\pi_\theta(p_w|u)}{\pi_{\mathrm{ref}}(p_w|u)} - \beta \log \frac{\pi_\theta(p_l|u)}{\pi_{\mathrm{ref}}(p_l|u)} \tag{19}
$$

The CW-DPO loss is $\mathcal{L}(\theta) = -\mathbb{E}[\omega \log \sigma(u_\theta)]$. Its gradient is:

$$
\nabla_\theta \mathcal{L}(\theta) = -\mathbb{E}\left[\omega \cdot \frac{1}{\sigma(u_\theta)} \cdot \sigma'(u_\theta) \cdot \nabla_\theta u_\theta\right] \tag{20}
$$

Using the identity $\sigma'(x) = \sigma(x)(1 - \sigma(x))$, we simplify:

$$
\nabla_\theta \mathcal{L}(\theta) = -\mathbb{E}\left[\omega \cdot (1 - \sigma(u_\theta)) \cdot \nabla_\theta u_\theta\right] \tag{21}
$$

Since $1 - \sigma(x) = \sigma(-x)$, and letting $\hat{r}_\theta(u,p) = \beta \log \frac{\pi_\theta(p|u)}{\pi_{\mathrm{ref}}(p|u)}$, the gradient becomes:

$$
\nabla_\theta \mathcal{L}(\theta) = -\mathbb{E}\left[\underbrace{\omega}_{\text{Coordination Weight}} \cdot \underbrace{\sigma(\hat{r}_\theta(u,p_l) - \hat{r}_\theta(u,p_w))}_{\text{Relative Error Weight}} \cdot \nabla_\theta(\hat{r}_\theta(u,p_w) - \hat{r}_\theta(u,p_l))\right] \tag{22}
$$

This formulation demonstrates that the gradient update is scaled by two adaptive factors: *Relative Error Weight*, which modulates the gradient magnitude based on the degree of preference misalignment, and *Coordination Weight*, which amplifies updates for plan pairs with larger gaps in planning executability.

### D.3. Gradient Analysis of CW-KTO

Following the Equation 12, the coordination-weighted KTO objective combines supervised fine-tuning with preference learning:

$$\mathcal{L}_{\text{CW-KTO}}(\pi_e, \pi_{\text{ref}}) = \underbrace{-\mathbb{E}_{(x,y)\sim\mathcal{D}_{\text{des}}}[c_\tau \log \pi_e(y \mid x)]}_{\mathcal{L}_{\text{SFT}}} + \underbrace{\mathbb{E}_{(x,y)\sim\mathcal{D}}[\lambda_y - v(x,y)]}_{\mathcal{L}_{\text{KTO}}}, \tag{23}$$

where $r_e(x,y) = \log \frac{\pi_e(y|x)}{\pi_{\text{ref}}(y|x)}$, $z_0 = \text{KL}\big(\pi_e(\cdot \mid x)\|\pi_{\text{ref}}(\cdot \mid x)\big)$, and

$$v(x,y) = \begin{cases} \lambda_D \sigma\big(\beta(r_e - z_0)\big) & y \sim y_{\text{desirable}}, \\ \lambda_U \sigma\big(\beta(z_0 - r_e)\big) & y \sim y_{\text{undesirable}}. \end{cases} \tag{24}$$

The gradient of SFT Component is:

$$\nabla_\theta \mathcal{L}_{\text{SFT}} = -\mathbb{E}_{(x,y)\sim\mathcal{D}_{\text{desirable}}}[c_\tau \cdot \nabla_\theta \log \pi_e(y \mid x)]. \tag{25}$$

$\lambda_y$ denotes $\lambda_D$ ($\lambda_U$) when y is desirable (undesirable), the gradient of KTO Component is

$$\nabla_\theta \mathcal{L}_{\text{KTO}} = -\mathbb{E}_{(x,y)\sim\mathcal{D}}[\nabla_\theta v(x,y)]. \tag{26}$$

For desirable samples ($y \sim y_{\text{desirable}}$), apply the chain rule:

$$\begin{aligned} \nabla_\theta v(x,y) &= \lambda_D \cdot \sigma'\big(\beta(r_e - z_0)\big) \cdot \beta \cdot \nabla_\theta(r_e - z_0) \\ &= \lambda_D \beta \cdot \sigma(\beta\Delta_D)\big(1 - \sigma(\beta\Delta_D)\big) \cdot \big(\nabla_\theta r_e - \nabla_\theta z_0\big), \end{aligned} \tag{27}$$

where $\Delta_D = r_e - z_0$ and $\sigma'(z) = \sigma(z)(1 - \sigma(z))$, and $\nabla_\theta r_e = \nabla_\theta \log \pi_e(y \mid x)$ since $\pi_{\text{ref}}$ is fixed.

For undesirable samples ($y \sim y_{\text{undesirable}}$):

$$\begin{aligned} \nabla_\theta v(x,y) &= \lambda_U \cdot \sigma'\big(\beta(z_0 - r_e)\big) \cdot \beta \cdot \nabla_\theta(z_0 - r_e) \\ &= -\lambda_U \beta \cdot \sigma(\beta\Delta_U)\big(1 - \sigma(\beta\Delta_U)\big) \cdot \big(\nabla_\theta r_e - \nabla_\theta z_0\big), \end{aligned} \tag{28}$$

where $\Delta_U = z_0 - r_e$.

Using the policy gradient identity for expectations under $\pi_e$:

$$\nabla_\theta z_0 = \nabla_\theta \mathbb{E}_{y'\sim\pi_e(\cdot|x)}[r_e(x,y')] = \mathbb{E}_{y'\sim\pi_e(\cdot|x)}[\nabla_\theta \log \pi_e(y' \mid x) \cdot r_e(x,y')]. \tag{29}$$

Substituting both components:

$$\begin{aligned} \nabla_\theta \mathcal{L}_{\text{CW-KTO}} = &-\mathbb{E}_{\mathcal{D}_{\text{desirable}}}[c_\tau \nabla_\theta \log \pi_e(y \mid x)] \\ &- \mathbb{E}_{\mathcal{D}_{\text{desirable}}}[\lambda_D \beta \sigma(\beta\Delta_D)(1 - \sigma(\beta\Delta_D))(\nabla_\theta \log \pi_e(y \mid x) - \nabla_\theta z_0)] \\ &+ \mathbb{E}_{\mathcal{D}_{\text{undesirable}}}[\lambda_U \beta \sigma(\beta\Delta_U)(1 - \sigma(\beta\Delta_U))(\nabla_\theta \log \pi_e(y \mid x) - \nabla_\theta z_0)]. \end{aligned} \tag{30}$$

Grouping terms containing $\nabla_\theta \log \pi_e(y \mid x)$ (direction) and $\nabla_\theta z_0$ (KL constraint):

$$\nabla_\theta \mathcal{L}_{\text{CW-KTO}} = \underbrace{\mathbb{E}_{\mathcal{D}}[w_{\text{dir}}(x,y) \cdot \nabla_\theta \log \pi_e(y \mid x)]}_{\text{Direction term}} + \underbrace{\mathbb{E}_{\mathcal{D}}[w_{\text{KL}}(x,y) \cdot \nabla_\theta z_0]}_{\text{KL regularization term}}, \tag{31}$$

where the sample-dependent weights are:

$$w_{\text{dir}}(x,y) = \begin{cases} -c_\tau - \lambda_D \beta \, \sigma(\beta\Delta_D)\big(1 - \sigma(\beta\Delta_D)\big) & y \sim y_{\text{desirable}}, \\ \lambda_U \beta \, \sigma(\beta\Delta_U)\big(1 - \sigma(\beta\Delta_U)\big) & y \sim y_{\text{undesirable}}, \end{cases}$$

$$w_{\mathrm{KL}}(x, y) = \begin{cases} \lambda_D \beta \, \sigma(\beta \Delta_D)\big(1 - \sigma(\beta \Delta_D)\big) & y \sim y_{\mathrm{desirable}}, \\ -\lambda_U \beta \, \sigma(\beta \Delta_U)\big(1 - \sigma(\beta \Delta_U)\big) & y \sim y_{\mathrm{undesirable}}. \end{cases}$$

The direction term drives policy updates toward high-reward actions while explicitly weighting desirable trajectories by coordination score $c_\tau \in [0, 1]$. Higher $c_\tau$ amplifies gradient magnitude for well-coordinated executions. The KL regularization term constrains policy deviation from $\pi_{\mathrm{ref}}$. Positive weights for desirable samples encourage controlled policy improvement, while negative weights for undesirable samples pull the policy back toward reference behavior. Coordination awareness enters explicitly through $c_\tau$ in $w_{\mathrm{dir}}$ and implicitly through preference pair selection (maximizing $c_{\tau_d} - c_{\tau_u}$), ensuring optimization prioritizes actions that both complete tasks and faithfully execute prescribed plans.

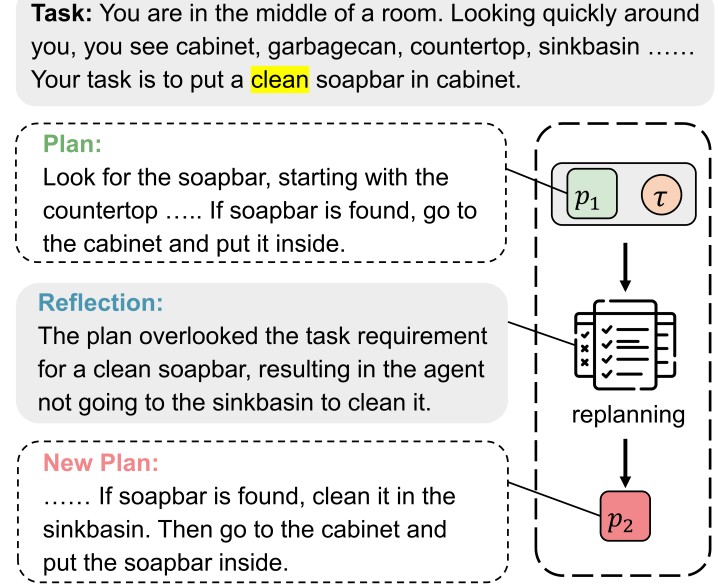

*Figure 10.* An example of agent self-replanning.

# E. Analysis of Computational Overhead

To address concerns regarding framework efficiency, we conducted a comprehensive analysis of its complexity from both theoretical and empirical perspectives.

**Theoretical Complexity.** The primary computational overhead arises from the evaluation phase of Self-Refining MCTS. The total time complexity is approximated as $T_{\mathrm{total}} \approx O(M \cdot n \cdot L_{\mathrm{exec}} \cdot T_{\mathrm{LLM}})$, where $L_{\mathrm{exec}}$ denotes the maximum step number and $T_{\mathrm{LLM}}$ represents the latency for a single LLM-environment interaction. Crucially, this formulation indicates that costs scale linearly with respect to $M$ and $n$, rather than exponentially, thereby guaranteeing controllable and predictable overhead as the system scales.

**Empirical Efficiency.** We measured the time consumption of each stage for a single agent optimization iteration using Qwen2.5-7B on four NVIDIA A100 GPUs. As detailed in Table 13, *Data Collection* constitutes the dominant cost but remains within practical bounds. *Coordination Assessment* incurs minimal overhead, while *Agent Optimization* consistently requires approximately 2 hours.

*Table 13.* Time consumption of each stage for a single agent optimization iteration.

| Dataset | Data Collection | Coordination Assessment | Agent Optimization |
|---------|-----------------|-------------------------|--------------------|
| AlfWorld | 2 days, 12 hours | 30 minutes | 2 hours |
| ScienceWorld | 3 days, 3 hours | 40 minutes | 2 hours |

*Table 14.* An example of alignment between the plan and execution trajectories. Two successful trajectories $\tau_A$ and $\tau_B$ for task $u$. Actions are denoted as $a_i^A$ or $a_i^B$, observations follow each action. Plan steps $s_j$ are shown on the left.

| $u$ | $p$ | $\tau_A$ | $\tau_B$ |
|---|---|---|---|
| put a clean cloth in drawer | $s_1$: go to possible location of cloth (e.g. handtowelholder, countertop ...) | $a_1^A$: **Action**: go to handtowelholder. **Observation**: you see nothing | $a_1^B = a_1^A$ |
| | $s_2$: once cloth is found, take it | $a_2^A$: **Action**: go to countertop. **Observation**: you see a cloth | $a_2^B = a_2^A$ |
| | $s_3$: clean the cloth in the sinkbasin | $a_3^A$: **Action**: take cloth from countertop. **Observation**: you pick up the cloth from the countertop | $a_3^B = a_3^A$ |
| | $s_4$: put the cloth in drawer | $a_4^A$: **Action**: go to sinkbasin. **Observation**: you see nothing | $a_4^B$: **Action**: go to drawer. **Observation**: the drawer is open. In it, you see nothing. |
| | | $a_5^A$: **Action**: clean cloth with sinkbasin. **Observation**: you clean the cloth using the sinkbasin | $a_5^B$: **Action**: put cloth in/on drawer. **Observation**: you put cloth in/on drawer |
| | | $a_6^A$: **Action**: go to drawer. **Observation**: the drawer is open. In it, you see nothing. | $a_6^B$: **Action**: take cloth from drawer. **Observation**: you pick up the cloth from the drawer |
| | | $a_7^A$: **Action**: put cloth in/on drawer. **Observation**: you have successfully completed the task | $a_7^B = a_4^A$ |
| | | | $a_8^B = a_5^A$ $a_9^B = a_6^A$ $a_{10}^B = a_7^A$ |

# F. Case Study of CoPE

## F.1. Case of Self-Replanning

As shown in Figure 10, the task requires the soapbar to be clean, which was overlooked in the original plan $p_1$. LLM agent performs the self-refine based on $p_1$ and its failed trajectory $\tau$ to generates a new plan $p_2$ that resolves this issue.

## F.2. Case of Execution Adherence Alignment

Regarding the execution adherence metric, to mitigate noise from redundant actions during semantic alignment, we designed **Algorithm 1** to finely align each planning step with its corresponding execution action. To illustrate the effectiveness of this dynamic programming algorithm, we present an example involving two successful trajectories $\tau_A$ and $\tau_B$ in Table 14. Notably, in $\tau_B$, the agent initially deviates from the plan by putting an uncleaned cloth directly into the drawer at step $a_5^B$, before later recovering.

We compute similarity matrices $M_A$ and $M_B$ between plan steps and executed actions. Note that $M_B$ has more columns due to extra actions in $\tau_B$:

$$M_A = \begin{bmatrix} 0.86 & 0.82 & 0.15 & 0.79 & 0.21 & 0.82 & 0.16 \\ 0.21 & 0.19 & 0.86 & 0.13 & 0.22 & 0.14 & 0.23 \\ 0.21 & 0.16 & 0.19 & 0.25 & 0.89 & 0.16 & 0.19 \\ 0.13 & 0.13 & 0.15 & 0.13 & 0.15 & 0.20 & 0.92 \end{bmatrix}, \quad M_B = \begin{bmatrix} 0.86 & 0.82 & 0.15 & 0.82 & 0.16 & 0.12 & 0.79 & 0.21 & 0.82 & 0.16 \\ 0.21 & 0.19 & 0.86 & 0.14 & 0.23 & 0.76 & 0.13 & 0.22 & 0.14 & 0.23 \\ 0.21 & 0.16 & 0.19 & 0.16 & 0.19 & 0.14 & 0.25 & 0.89 & 0.16 & 0.19 \\ 0.13 & 0.13 & 0.15 & 0.20 & 0.92 & 0.23 & 0.13 & 0.15 & 0.20 & 0.92 \end{bmatrix}$$

Using dynamic programming, we derive cumulative alignment scores $\mathrm{DP}_A$ and $\mathrm{DP}_B$, where boxed values indicate optimal matches:

$$\mathrm{DP}_A = \begin{bmatrix} 0.86 & \boxed{0.82} & 0.72 & 0.79 & 0.69 & 0.82 & 0.72 \\ 0.46 & 1.05 & \boxed{1.68} & 1.58 & 1.48 & 1.38 & 1.28 \\ 0.82 & 1.29 & 1.84 & \boxed{2.63} & 2.53 & 2.43 & 2.33 \\ 0.42 & 0.95 & 1.44 & 2.23 & 2.78 & 2.73 & \boxed{3.35} \end{bmatrix}, \quad \mathrm{DP}_B = \begin{bmatrix} 0.86 & \boxed{0.82} & 0.72 & 0.82 & 0.72 & 0.62 & 0.79 & 0.69 & 0.82 & 0.72 \\ 0.46 & 1.05 & 1.68 & 1.58 & 1.48 & \boxed{1.48} & 1.38 & 1.28 & 1.18 & 1.08 \\ 0.21 & 0.65 & 1.28 & 1.84 & 1.77 & 1.67 & 1.73 & \boxed{2.27} & 2.17 & 2.07 \\ 0.13 & 0.34 & 0.88 & 1.48 & 2.76 & 2.66 & 2.55 & 2.45 & 2.47 & \boxed{3.09} \end{bmatrix}$$

The final alignment score is computed as the last element of the DP matrix divided by the number of plan steps (4):

$$\text{Score}_A = \frac{3.35}{4} = 0.8375, \quad \text{Score}_B = \frac{3.09}{4} = 0.7725.$$

Although $\tau_B$ eventually recovers to the correct path after $a_6^B$, the premature placement of an uncleaned cloth into the drawer at $a_5^B$ introduces a penalty in execution adherence. This demonstrates that our metric is sensitive to minor deviations and can capture semantically misaligned successes.

# G. Prompts for CoPE

In this section, we provide all prompts used in CoPE. Placeholders enclosed in curly braces (e.g., {instruction}, {task}, {demonstration}) are replaced with actual text at runtime.

---

**Plan Generation Prompt for AlfWorld**

```
Please generate a step-by-step plan for a house holding task.  The output format is
as follows:
<plan>
1.  [Step 1]
2.  [Step 2]
3.  ...
</plan>

The available actions are:
1.  go to recep
2.  take obj from recep
3.  put obj in/on recep
4.  open recep
5.  close recep
6.  toggle obj recep
7.  clean obj with recep
8.  heat obj with recep
9.  cool obj with recep
where obj and recep correspond to objects and receptacles.

Below is one demonstration for generating a plan:
{demonstration}

Now, it's your turn, here is the task:
<task>
{instruction}
</task>
```

---

**Plan Generation Prompt for ScienceWorld**

```
You are a helpful assistant to do some scientific experiment in an environment.
Please generate a step-by-step plan for a scientific task.
In the environment, there are several rooms:  kitchen, foundry, workshop, bathroom,
outside, living room, bedroom, greenhouse, art studio, hallway.

You should explore the environment and find the items you need to complete the
experiment.
You can teleport to any room in one step.
All containers in the environment have already been opened, you can directly get
items from the containers.

The available actions are:
open OBJ: open a container
close OBJ: close a container
activate OBJ: activate a device
deactivate OBJ: deactivate a device
connect OBJ to OBJ: connect electrical components
disconnect OBJ: disconnect electrical components
use OBJ [on OBJ]: use a device/item
look around:  describe the current room
examine OBJ: describe an object in detail
look at OBJ: describe a container's contents
read OBJ: read a note or book
```

```
move OBJ to OBJ: move an object to a container
pick up OBJ: move an object to the inventory
pour OBJ into OBJ: pour a liquid into a container
mix OBJ: chemically mix a container
teleport to LOC: teleport to a specific room
focus on OBJ: signal intent on a task object
wait:  task no action for 10 steps
wait1:  task no action for a step

The generated plan should be written in the following format:
<plan>
Step 1:  ...
Step 2:  ...
...
</plan>

Below is two demonstration for generating a plan:
{demonstrations}

Now, it's your turn, here is the task:
<task>
{instruction}
</task>
```

**Execution Prompt for AlfWorld**

```
Interact with a household to solve a task.  Imagine you are an intelligent agent
in a household environment and your target is to perform actions to complete the
task goal.  At the beginning of your interactions, you will be given the detailed
description of the current environment and your goal to accomplish.
For each of your turn, you will be given the observation of the last turn.  You
should choose from two actions:  "Thought" or "Action".  If you choose "Thought",
you should first think about the current condition and plan for your future actions,
and then output your action in this turn.  Your output must strictly follow this
format:"Thought:  your thoughts.  Action:  your next action"; If you choose "Action",
you should directly output the action in this turn.  Your output must strictly
follow this format:"Action:  your next action".
The available actions are:
1.  go to recep
2.  take obj from recep
3.  put obj in/on recep
4.  open recep
5.  close recep
6.  toggle obj recep
7.  clean obj with recep
8.  heat obj with recep
9.  cool obj with recep
where obj and recep correspond to objects and receptacles.
After your each turn, the environment will give you immediate feedback based on
which you plan your next few steps.  if the enviroment output "Nothing happened",
that means the previous action is invalid and you should try more options.
Reminder:
1.  The action must be chosen from the given available actions.  Any actions except
provided available actions will be regarded as illegal.
2.  Think when necessary, try to act directly more in the process.

Now, it's your turn and here is the task.
{task}

This plan maybe helpful for you to complete the task:
```

```
{plan}
```

---

**Execution Prompt for ScienceWorld**

```
You are a helpful assistant to do some scientific experiment in an environment.
In the environment, there are several rooms:  kitchen, foundry, workshop, bathroom,
outside, living room, bedroom, greenhouse, art studio, hallway
You should explore the environment and find the items you need to complete the
experiment.
You can teleport to any room in one step.
All containers in the environment have already been opened, you can directly get
items from the containers.
For each of your turn, you will be given the observation of the last turn.  You
should choose from two actions:  "Thought" or "Action".  If you choose "Thought",
you should first think about the current condition and plan for your future actions,
and then output your action in this turn.  Your output must strictly follow this
format:"Thought:  your thoughts.  Action:  your next action"; If you choose "Action",
you should directly output the action in this turn.  Your output must strictly
follow this format:"Action:  your next action".  Remember that you can only output
one "Action:" in per response.

The available actions are:
open OBJ: open a container
close OBJ: close a container
activate OBJ: activate a device
deactivate OBJ: deactivate a device
connect OBJ to OBJ: connect electrical components
disconnect OBJ: disconnect electrical components
use OBJ [on OBJ]: use a device/item
look around:  describe the current room
examine OBJ: describe an object in detail
look at OBJ: describe a container's contents
read OBJ: read a note or book
move OBJ to OBJ: move an object to a container
pick up OBJ: move an object to the inventory
pour OBJ into OBJ: pour a liquid into a container
mix OBJ: chemically mix a container
teleport to LOC: teleport to a specific room
focus on OBJ: signal intent on a task object
wait:  task no action for 10 steps
wait1:  task no action for a step

Now, it's your turn and here is the task.
{task}

This plan maybe helpful for you to complete the task:
{plan}
```

---

**Prompt for Replanning**

```
Generate an improved plan by analyzing the task description, input instruction,
prior plan (sub-steps), and failed trajectory.  Focus on addressing root causes of
failure while preserving valid components of the prior plan.

Task:
```

```
<description>
{task description}
</description>

Input Instruction:
<instruction>
{instruction}
</instruction>

Prior plan:
<plan>
{plan}
</plan>

Failure Trajectory:
<trajectory>
{trajectory}
</trajectory>

Please generate a new plan that includes sub-steps.  The new plan should not include
specific thoughts or actions and should be as concise as possible.  The output
format is as follows:
<reflection>
Analysis of Error Causes
</reflection>

<plan>
1.  [Step 1]
2.  [Step 2]
3.  ...
</plan>
```

**Prompt for Reconstruction**

```
Please generate a step-by-step meta plan for a scientific task:
<task>
{task}
</task>

<description>
{task description}
</description>

Below is the standard and detailed procedure for solving this task:
<trajectory>
{successful trajectory}
</trajectory>

You need to conclude abstract steps as a plan, which can be used to solve similar
tasks in the future.
The plan should be a commonly-reused routine of the tasks.
The generated meta plan should be written in the following format:
<plan>
Step 1:  ...
Step 2:  ...
...
</plan>
```

**Prompt for Plan Correctness and Executability Assessment**

```
You are a professional guidance evaluator.  Please critically assess the provided
task plan for the given task along the following three dimensions.  Base your
evaluation on the task description, the current execution context (including the
execution step index and, if available, accumulated observations), and the plan
itself.

### Evaluation Dimensions:

1.  **Correctness**:
To what extent does the task plan align with the task requirements and lead the
agent toward successful task completion, considering the environment's feedback on
actions taken under this plan?

2.  **Executability**:
Is the plan clear, logically sequenced, and composed of actionable, reasonable steps
that an agent can reliably follow without ambiguity?

3.  **Standardization**:
Does the plan conform to a consistent, well-structured, and standardized format
(e.g., numbered steps, imperative verbs, object references with identifiers)?

### Input Context:

- **Task**:
task

- **Plan**:
plan

### Output Instructions:

Provide your evaluation in strict JSON format as shown below.  Assign an integer
score from 1 to 5 for each dimension (1 = poor, 5 = excellent), and include a
concise, evidence-based justification for each score.

```json

"correctness_score":  xxx,
"correctness_reason":  "...",
"executability_score":  xxx,
"executability_reason":  "...",
"standardization_score":  xxx,
"standardization_reason":  "..."

```
```

**Prompt for Execution Adherence Assessment**

```
You are an expert in agent tasks.  You are tasked with evaluating the agent's
execution of a given task plan.  Specifically, you are to assess the degree of
compliance between the agent's actions and the strategic guidance outlined in the
task plan.  Rate it from 0 to 2 points, and explain the reason.

**Scoring Criteria:**

- **2 Points**:
The agent's execution strictly adheres to the guidance provided in the task plan.
All actions are logically aligned with the plan's objectives and are carried out as
instructed.
```

```
- **1 Point**:
 The agent's execution demonstrates partial alignment with the task plan, allowing
 for minor deviations.  For example, if the plan suggests using multiple tools, the
 agent may use at least one relevant tool, as long as it does not contradict the
 overall guidance.

 - **0 Points**:
 The agent's execution departs from or contradicts the task plan, or contains garbled
 characters, format errors, disordered steps, or irrelevant information.

 **Input Context:**

 - **Task**:
 task

 - **Plan**:
 plan

 - **Agent-Environment Interaction**:
 accumulated context

 **Output Format:**
 Provide your evaluation in strict JSON format as shown below.  Do not include any
 additional text or explanation outside the JSON.

 ```json

 "score":  xxx,
 "reason":  "..."

 ```
```

*Table 15.* Action Vocabulary and Descriptions in AlfWorld and ScienceWorld. (LOC: location; OBJ: object; RECEP: receptacle.)

| Dataset | Action | Description |
|---|---|---|
| AlfWorld | goto LOC | move to a location and observe its contents |
| | take OBJ from RECEP | pick up an object from a container |
| | put OBJ in/on RECEP | place an object onto a surface or container |
| | open RECEP | open a container to reveal its contents |
| | close RECEP | close a container |
| | toggle OBJ | turn on/off a device |
| | heat OBJ with RECEP | heat an object using a heating device |
| | cool OBJ with RECEP | cool an object using a cooling device |
| | clean OBJ with RECEP | clean an object using a cleaning tool |
| | examine RECEP | inspect an object or container for details |
| ScienceWorld | open/close OBJ | open/close a container |
| | de/activate OBJ | activate/deactivate a device |
| | connect OBJ to OBJ | connect electrical components |
| | disconnect OBJ | disconnect electrical components |
| | use OBJ [on OBJ] | use a device/item |
| | look around | describe the current room |
| | look at OBJ | describe an object in detail |
| | look in OBJ | describe a container's contents |
| | read OBJ | read a note or book |
| | move OBJ to OBJ | move an object to a container |
| | pick up OBJ | move an object to the inventory |
| | put down OBJ | drop an inventory item |
| | pour OBJ into OBJ | pour a liquid into a container |
| | dunk OBJ into OBJ | dunk a container into a liquid |
| | mix OBJ | chemically mix a container |
| | go to LOC | move to a new location |
| | teleport to LOC | teleport to a specific room |
| | eat OBJ | eat a food |
| | flush OBJ | flush a toilet |
| | focus on OBJ | signal intent on a task object |
| | wait [DURATION] | take no action for some duration |
| | task | describe current task |
| | inventory | list agent's inventory |

