# OpenReview forum: "CoPE: A Framework for Optimizing Coordination between Planning and Execution in LLM-based Agents"
_ICML.cc/2026/Conference — ICML 2026 regular_

### Official Review · Reviewer_kzLq · 2026-02-20

**Soundness:** 3
**Presentation:** 3
**Significance:** 3
**Originality:** 3
**Overall Recommendation:** 4
**Confidence:** 2

**Summary:**

This paper targets at the suboptimal plan and  plan-execute inconsistency patterns of existing agentic LLMs on long-horizon tasks. It proposes CoPE to integrate planning-execution coordination for agent optimization where the finegrained filtering/weighting criteria are applied for trajectory selection and the self-refining MCTS is involved to boost refine the plans through interactions. Results on alfworld and scienceworld confirm the efficiency and effectiveness of interaction trajectorie patterns cultivated by CoPE.

**Compliance With Llm Reviewing Policy:**

Affirmed.

**Final Justification:**

The authors resolved most of my concerns.

**Key Questions For Authors:**

- Does the proposed method ensure a fair comparison (correct me if the reviewer misunderstands) where the alternative optimization methods all use the same training data (e.g., filtered trajectoris after sec.4.2 & 4.3 for SFT)?
- The authors are encouraged to investigate if the coordination-based reweighting can be applied on techniques such as RFT/ETO/MPO/vanilla DPO?
- Can the authors provide performance on more sophisticated tasks such as retrieval enhanced QA (Single-hop/Multi-hop QA/) and deep search (e.g., gaia) for demonstrations of the generalizability?
- In Appendix B, why not directly use the LLM-as-a-Judge for scoring the plan-execution alignment?

**Limitations:**

yes

**Strengths And Weaknesses:**

### Summary Of Strengths
- The paper is generally well-written with good illustrations.
- The core components of data collection, coordination assessment, agent optimization are intuitive and straightforward.


### Summary Of Weaknesses
- The proposed method relies heavily on the data collection effectiveness with self-refining MCTS. However, it remains unknown whether such meticulous refining progress can generalize to complicated agent scenarios such as coding. In addition, the reviewer did not fully understand whether the previous, initial suboptimal plans (turns in each trajectory) are involved in training (e.g., loss computation and policy update).
- The coordination assessment (e.g., plan-executability and execution adherence assessment) plays a critical rule in discriminating trajectories with alignment score. However, the paper did not provide detailed analysis on the accuracy/precision of such assessment. It therefore remains unknown how robust the method is and how such assessment accuracy affects DP-based plan-execution alignment scoring.

---

> ### Author Rebuttal · Authors · 2026-03-29
>
> We thank the reviewer for raising important questions. We have incorporated the corresponding discussions into our manuscript.
>
> > W1.1: However, it remains unknown whether such meticulous refining progress can generalize to complicated agent scenarios such as coding.
>
> We sincerely appreciate the reviewer's concern about the generalizability. In fact, combining LLM’s refining mechanism with MCTS to collect data has been extensively validated across agent domains, including code generation[1], web navigation[2] and multi-hop QA[3].
>
>   [1] SRA-MCTS: Self-driven Reasoning Augmentation with Monte Carlo Tree Search for Code Generation (IJCAI 2025)
>
>   [2] ExACT: Teaching AI Agents to Explore with Reflective-MCTS and Exploratory Learning (ICLR 2025)
>
>   [3] Empowering Large Language Model Agent through Step-Level Self-Critique and Self-Training (SIGIR 2025)
>
> > W1.2: In addition, the reviewer did not fully understand whether the previous, initial suboptimal plans (turns in each trajectory) are involved in training.
>
> Initial suboptimal plans or trajectories can serve as positive or negative examples, if they satisfy the criteria of preference data construction in `Section 4.4`. In practice, the plan $p=\pi_p(u)$ and every action $a_t=\pi_e(u, p, \tau_{t-1})$ within the trajectory are involved in the loss computation.
>
> > W2: However, the paper did not provide detailed analysis on the accuracy/precision of such assessment. It therefore remains unknown how robust the method is and how such assessment accuracy affects DP-based plan-execution alignment scoring.
>
> We would like to clarify that our DP-based alignment is precisely used to compute the coordination score. The high rank correlation score of 0.83 between our method and LLM models (`Table 7, Appendix B`) demonstrates the accuracy of our assessment. In addition, the robustness of our method was discussed in detail in the response to **Reviewer vaE3** (see "W2")
>
> > Q1: Does the proposed method ensure a fair comparison where the alternative optimization methods all use the same training data?
>
> This is a fair comparison as sampling strategies are integral to the agent optimization framework; actually the comparison subject is the LLM agent constructed after multi-round data collection. Besides, the original papers of other baseline methods (e.g., WKM, ETO, MPO) were also compared in the same manner.
>
> To further address this concern, we trained the baselines with the **same training data** collected from CoPE's final iteration on AlfWorld. However, for baselines like KnowAgent, WKM and PiplotRL, our data are incompatible with their inference paradigms. The table below shows that while other methods exhibit slight gains, our CoPE maintains significantly superior performance.
>
>   |Model|Method|Seen|Unseen|
>   |-|-|-|-|
>   | Qwen2.5-7B|RFT|71.26|72.79|
>   | Qwen2.5-7B|ETO|73.43|74.29|
>   | Qwen2.5-7B|MPO|80.46|81.34|
>   | Qwen2.5-7B|AGENTEVOL|82.74|82.06|
>   | Qwen2.5-7B|CoPE (Our)|**88.00**|**86.27**|
>
> > Q2: If the coordination-based reweighting can be applied on techniques such as RFT/ETO/MPO/vanilla DPO?
>
> We thank the reviewer for the opportunity to clarify it. Our planning agent's optimization method, CW-DPO, is an extension of vanilla DPO that incorporates coordination-based reweighting. This reweighting mechanism is compatible with the model training method. Therefore, it is applicable to these techniques, including RFT, ETO, and MPO.
>
> > Q3: Can the authors provide performance on more sophisticated tasks such as retrieval enhanced QA (Single-hop/Multi-hop QA/) and deep search (e.g., gaia) for demonstrations of the generalizability?
>
> We appreciate the suggestion. However, GAIA is an evaluation benchmark with limited questions (~466) and private test answers, making it unsuitable for training-based evaluation. Instead, we conducted experiments on **2WikiMultihopQA** [4] for multi-hop QA. Due to character limits, the results are detailed in **our response to Reviewer c2iy** (see "W2"), demonstrating CoPE's significant improvements and validating its generalizability.
>
>   [4] Constructing a Multi‑hop QA Dataset for Comprehensive Evaluation of Reasoning Steps (COLING 2020)
>
> > Q4: Why not directly use the LLM-as-a-Judge for scoring the plan-execution alignment?
>
> We thank the reviewer for questioning the scoring method. Prior research [5][6][7] has shown that scalar scores generated by "LLM-as-a-Judge" exhibit high variance, making them unsuitable for building preference data. In practice, our metrics consistently produce a stable ranking, whereas the LLM usually reverses samples' preference across runs (e.g. scoring A>B in one trial and B>A in another). Such reversals introduce severe noise that destabilizes agent optimization.
>
>   [5] TrustJudge: Inconsistencies of LLM-as-a-Judge and How to Alleviate Them (ICLR 2026)
>
>   [6] Judging LLM-as-a-Judge with MT-Bench and Chatbot Arena (NeurIPS 2023)
>
>   [7] Rating Roulette: Self-Inconsistency in LLM-As-A-Judge Frameworks (EMNLP 2025)

---

> > ### Author Rebuttal · Reviewer_kzLq · 2026-04-03
> >
> > Thank you for the response. The concerns are resolved and the new experiments and analyses are encouraged to be added into the manuscript for improvement. The reviewer would increase the score to weak accept.

---

> > > ### Author Response · Authors · 2026-04-04
> > >
> > > We sincerely appreciate the reviewer's insightful comments and the increased score. We will incorporate the new results and discussion into our revision.

---

### Official Review · Reviewer_vaE3 · 2026-03-09

**Soundness:** 2
**Presentation:** 2
**Significance:** 3
**Originality:** 2
**Overall Recommendation:** 4
**Confidence:** 3

**Summary:**

This paper studies the planning–execution coordination problem in LLM-based agents for long-horizon tasks. It proposes the CoPE framework, which collects data via self-improving MCTS, evaluates coordination based on plan feasibility and execution consistency, and uses the coordination score to weight training samples. Experiments on AlfWorld and ScienceWorld show improvements over several baselines in task performance.

**Compliance With Llm Reviewing Policy:**

Affirmed.

**Final Justification:**

The Authors' response has addressed all of my main questions:

– addressed generalization concerns and validated the coordination score’s heuristic components via additional experiments, hyperparameter analysis, ablation studies, and correlation with LLM-as-a-Judge;

– supplemented computational overhead analysis, clarified training stability, identified the minimal rollout number, and explained metric misjudgment cases and trajectory distinction intuition.

One improvement is still desirable: briefly including a concise snippet of the qualitative example for the coordination score’s trajectory distinction in the paper would enhance clarity and make the argument more self-contained.

With this minor adjustment, the paper would be even more convincing. In my opinion, the current version already meets the bar for acceptance.

**Key Questions For Authors:**

1. How stable is the coordination score during training? For example, does the weighting scheme lead to training instability or large variance across runs?
2. How sensitive is the framework to the number of executions per plan during coordination estimation? Is there a minimal number of rollouts that still provides reliable signals?
3. Can the authors provide more intuition or qualitative examples showing how the coordination score distinguishes between good and bad trajectories?

**Limitations:**

While the paper mentions some limitations, it could further discuss issues related to generalization to other agent domains, computational cost, and potential bias introduced by heuristic coordination metrics.

**Strengths And Weaknesses:**

Strengths:
1. The paper highlights the mismatch between planning and execution in LLM-based agents, which is a relevant issue in long-horizon environments and is not always explicitly addressed in reward-based optimization methods.
2. The method is evaluated on AlfWorld and ScienceWorld, where it shows improvements over several baselines.

Weaknesses:
1. Experiments are conducted only on AlfWorld and ScienceWorld, leaving the generalization of the approach to other agent domains (e.g., web agents, tool-use agents, coding agents) unclear.
2. The coordination score relies on heuristic components (rule-based action matching, embedding similarity, and dynamic programming alignment), whose robustness and reliability are not systematically validated.
3. The framework requires multiple rollouts per plan, MCTS-based data collection, and trajectory alignment, but the paper lacks a detailed analysis of the resulting computational overhead.
4. The paper primarily reports improvements but does not analyze cases where the coordination metric may incorrectly score useful trajectories.

---

> ### Author Rebuttal · Authors · 2026-03-29
>
> We sincerely appreciate the reviewer's insightful comments. We have revised our manuscript to incorporate all the suggested changes.
>
> > W1: Experiments are conducted only on AlfWorld and ScienceWorld, leaving the generalization of the approach to other agent domains (e.g., web agents, tool-use agents, coding agents) unclear.
>
> We thank the reviewer’s for the comment regarding the scope of evaluation. To address the concern regarding generalization to other domains, we conducted additional experiments on **2WikiMultihopQA** [1], a challenging multi-hop QA task, where agents iteratively call a search engine tool to retrieve factual evidence. The table below shows that CoPE demonstrates significant performance improvements.
>
>   |Model|Method|Reward|
>   |-|-|-|
>   |GPT-4o|ReAct|57.31|
>   |Deepseek-Chat|ReAct|61.72|
>   |Gemini2.5-Pro|ReAct|51.60|
>   |Qwen2.5-7B|ReAct|40.37|
>   |Qwen2.5-7B|ETO|52.39|
>   |Qwen2.5-7B|RFT|49.76|
>   |Qwen2.5-7B|AGENTEVOL|54.35|
>   | Qwen2.5-7B|CoPE (Our)|**67.74**|
>
>   [1] Constructing a Multi‑hop QA Dataset for Comprehensive Evaluation of Reasoning Steps (COLING 2020)
>
> > W2: The coordination score relies on heuristic components whose robustness and reliability are not systematically validated.
>
> We thank the reviewer for this insightful comment. We have validated the **robustness** of our heuristic components through the following experiments:
> 1. **Hyperparameter Robustness.** As detailed in Figure 8 (`Appendix A, Page 13`), we conducted extensive analysis on hyperparameters `𝜆1` and `𝜆2` in dynamic programming. CoPE shows minimal performance variance across different 𝜆 settings while consistently surpassing the "w/o Coord-" baseline, demonstrating robust parameter stability.
> 2. **Ablation of Rule-Based Component.** In 2WikiMultihopQA, the rule-based component is inapplicable as only a single search tool is available. While CoPE remains effective by relying solely on embedding cosine similarity, demonstrating its inherent robustness.
>
> In addition, our metric’s **reliability** is evidenced by a strong Spearman correlation of 0.83 with LLM-as-a-Judge (`Appendix B, Table 7, Page 15`).
>
> > W3: The paper lacks a detailed analysis of the resulting computational overhead.
>
>  We thank the reviewer's for raising concerns about the computational overhead. Due to character limits, we present a detailed analysis of the time cost and complexity of each stage of the framework in **our response to Reviewer c2iy** (see the response to "W1"). We have added this discussion in our manuscript.
>
> > W4: The paper primarily reports improvements but does not analyze cases where the coordination metric may incorrectly score useful trajectories.
>
> In early optimization iterations, suboptimal plans can cause metric misjudgments. For example, consider the task "put some box on the sofa". A suboptimal plan contains a flawed step - "if the box is not found, look for another object that can be placed on the sofa".  Such plan could incorrectly assign high execution adherence scores to trajectories that follow this erroneous logic. However, most incorrectly scored trajectories will be filtered out based on task rewards.
>
> > Q1: How stable is the coordination score during training?
>
> We thank the reviewer for the attention to our training stability. We trained five Qwen2.5-7B Execution Agents with different seeds and saved their models. The table below shows that their average rewards exhibit negligible variance across five independent runs on Alfworld Seen, which demonstrates the training stability of our weighting scheme.
>
>   |model 1|model 2|model 3|model 4|model 5|
>   |-|-|-|-|-|
>   |88.00±0.56|88.12±0.83|87.45±0.32|86.98±0.96|87.43±0.51|
>
> > Q2: How sensitive is the framework to the number of executions per plan during coordination estimation? Is there a minimal number of rollouts that still provides reliable signals?
>
> We added experiments on AlfWorld with Qwen2.5-7B, evaluating 200 task samples with rollout number $n$ of 3 and 7. Following `Table 5 (Page 14)`, we calculated the coordination gap between successful and failed trajectories. The table below shows that the gap is clearly distinct at $n=3$ and widens with more rollouts. This suggests that $n=3$ is the minimum number for AlfWorld.
>
> |$n$|3|5|7|
> |-|-|-|-|
> |Consistency Gap|0.27|0.34|0.36|
> |Similarity Gap|0.08|0.15|0.16|
>
> > Q3: Can the authors provide more intuition or qualitative examples showing how the coordination score distinguishes between good and bad trajectories?
>
> We appreciate the request for more qualitative intuition. Due to character limits, we have provided a detailed quantitative example illustrating how the coordination score distinguishes good and bad trajectories in **our response to Reviewer MqPe** (see the response to "W1.2").
> Briefly, this example demonstrates that even if there are only one or two deviating actions, this can lead to a significant decrease in our coordination score. We hope this cross-reference provides the necessary clarity.

---

> > ### Author Rebuttal · Reviewer_vaE3 · 2026-04-03
> >
> > Thanks for your response. I have no questions. I will maintain the positive score.

---

> > > ### Author Response · Authors · 2026-04-05
> > >
> > > We sincerely appreciate the reviewer’s insightful comments, particularly regarding the generalization to broader agent domains and the robustness and intuition behind the coordination score. We have incorporated all relevant discussions into our revised manuscript. We sincerely appreciate the reviewer’s thoroughness and attention to detail, which reflect a high standard of professional rigor.
> > >
> > > Given that the concerns have been kindly marked as "Fully resolved", we would be grateful if the reviewer could consider revisiting the current score to reflect this updated assessment.

---

### Official Review · Reviewer_MqPe · 2026-03-10

**Soundness:** 4
**Presentation:** 4
**Significance:** 3
**Originality:** 3
**Overall Recommendation:** 5
**Confidence:** 4

**Summary:**

This paper introduces CoPE (A Framework for Optimizing Coordination between Planning and Execution in LLM-based Agents), a novel framework that explicitly integrates planning–execution coordination into the optimization of LLM-based agents. The proposed approach employs a Self-Refining Monte Carlo Tree Search (MCTS) algorithm to generate task plans and multiple execution trajectories through iterative interactions with the environment.

Large Language Models (LLMs) are further fine-tuned on domain-specific datasets, enabling the model to quantify and optimize the degree of coordination between planning and execution. This mechanism addresses the challenge of balancing long-horizon planning and multi-step execution in interactive real-world tasks, thereby improving the optimization of coordination weights between these two components. As a result, LLM-based agents are able to learn more effective strategies for both planning and execution.

Extensive experimental evaluations demonstrate that CoPE substantially enhances the coordination capability of intelligent agents. On benchmark tasks involving two long-horizon, multi-step scenarios, the proposed framework consistently outperforms state-of-the-art baseline methods, highlighting its effectiveness in improving coordinated decision-making within LLM-based agent systems.

**Compliance With Llm Reviewing Policy:**

Affirmed.

**Key Questions For Authors:**

1.	In the prompts presented in the appendix, several noticeable spelling errors appear. Specifically, the variable name {insturction} is repeatedly misspelled in the prompt templates on pages 21 and 22. The correct spelling should be {instruction}. Such typographical inconsistencies in variable names may cause confusion when reproducing the prompts or implementing the framework, and therefore should be carefully corrected for clarity and accuracy.
2.	CoPlanExec vs.\ CoPE: Although the framework is consistently referred to as \textbf{CoPE} throughout the paper, the legend in Figure 5 labels the corresponding curve as \textbf{CoPlanExec}. This inconsistency in terminology may lead to ambiguity regarding whether the two names refer to the same framework or represent different methods. A consistent naming convention should therefore be adopted across the manuscript and all figures to ensure clarity and avoid potential confusion for readers.

**Limitations:**

The proposed approach demonstrates strong performance in multi-step, long-horizon tasks. However, in short-term tasks, the misalignment between planning and execution does not necessarily correlate with task completion outcomes. As illustrated in Figure 3, the distribution of planning–execution coordination indicates that the Semantic Similarity between the Plan and the Execution Trajectory exhibits a substantial degree of overlap. This observation suggests that discrepancies in planning–execution coordination may exert limited influence on task success in shorter tasks and may also have minimal impact on the overall quality of the generated plans.

**Strengths And Weaknesses:**

Strengths

The paper identifies an interesting problem in LLM-based agents, namely the coordination between planning and execution. The motivation is clearly explained, and the authors provide intuitive examples showing how inconsistencies between plans and executions may affect training.
The proposed framework is also relatively easy to understand. By generating multiple trajectories and evaluating their coordination with the original plan, the method attempts to prioritize more reliable training data. In addition, the use of Monte Carlo Tree Search provides a systematic way to explore multiple candidate trajectories.
Finally, the experimental results show improvements over several baselines, suggesting that the proposed approach may be useful for improving agent performance.

Weaknesses

Although the framework is reasonable, several components rely on existing techniques such as Monte Carlo Tree Search and trajectory sampling. Therefore, the main contribution mainly lies in how these techniques are integrated into the proposed framework.
In addition, the paper provides limited discussion on how the coordination metric is designed. A more detailed explanation could help readers better understand why this metric effectively measures planning–execution consistency.
Finally, the experimental analysis could be further strengthened. For example, additional ablation studies could help clarify the contribution of different components in the framework.

---

> ### Author Rebuttal · Authors · 2026-03-29
>
> We sincerely appreciate the reviewer's comprehensive and meticulous review.
>
> > W1.1: Although the framework is reasonable, several components rely on existing techniques ... Therefore, the main contribution mainly lies in how these techniques are integrated into the proposed framework.
>
> We fully agree that MCTS and trajectory sampling are established techniques; however, our core contribution lies in proposing a novel coordination mechanism that effectively bridges the gap between planning and execution to address the challenges of long-horizon tasks.
>
> > W1.2: A more detailed explanation could help readers better understand why this metric effectively measures planning–execution consistency.
>
> To help the reviewer understand intuitively, we use an example to illustrate the effectiveness of our algorithm. The table below shows **two successful trajectories** $\tau_A$ and $\tau_B$. Notably, in $\tau_B$, the agent initially deviates from the plan at step $a^B_4$.
>
>   |$u$|$p$|$\tau_A$|$\tau_B$|
>   |-|-|-|-|
>   |put a clean cloth in drawer|$s_1$. go to possible location of cloth (e.g. handtowellholder, countertop ...)|$a^A_1$. **Action**: go to handtowellholder. **Observation**: you see nothing|$a^B_1$ = $a^A_1$|
>   ||$s_2$. once cloth is found, take it|$a^A_2$. **Action**: go to countertop. **Observation**: you see a cloth | $a^B_2$ = $a^A_2$|
>   ||$s_3$. clean the cloth in the sinkbasin| $a^A_3$. **Action**: take cloth from countertop. **Observation**: you pick up the cloth from the countertop|$a^B_3$ =$a^A_3$|
>   ||$s_4$. put the cloth in drawer|$a^A_4$. **Action**: go to sinkbasin. **Observation**: you see nothing|$a^B_4$. **Action**: go to drawer. **Observation**: the drawer is open. In it, you see nothing.|
>   ||| $a^A_5$. **Action**: clean cloth with sinkbasin. **Observation**: you clean the cloth using the sinkbasin|$a^B_5$. **Action**: put cloth in/on drawer. **Observation**: you put cloth in/on drawer |
>   ||| $a^A_6$. **Action**: go to drawer. **Observation**: the drawer is open. In it, you see nothing. | $a^B_6$. **Action**: take cloth from drawer. **Observation**: you pick up the cloth from the drawer |
>   ||| $a^A_7$. **Action**: put cloth in/on drawer. **Observation**: you have successfully completed the task | $a^B_7$ = $a^A_4$|
>   |||| $a^B_8$=$a^A_5$|
>   |||| $a^B_9$=$a^A_6$ |
>   |||| $a^B_{10}$=$a^A_7$|
>
>   We can obtain two similarity matrices $M_A$ and $M_B$ of these two trajectories:
> $$
>   M_A=\begin{bmatrix}
>   0.86&0.82&0.15&0.79&0.21&0.82&0.16\\\\
>   0.21&0.19&0.86&0.13&0.22&0.14&0.23\\\\
>   0.21&0.16&0.19&0.25&0.89&0.16&0.19\\\\
>   0.13&0.13&0.15&0.13&0.15&0.20&0.92
>   \end{bmatrix},
>   M_B=\begin{bmatrix}
>   0.86&0.82&0.15&0.82&0.16&0.12&0.79&0.21&0.82&0.16\\\\
>   0.21&0.19&0.86&0.14&0.23&0.76&0.13&0.22&0.14&0.23\\\\
>   0.21&0.16&0.19&0.16&0.19&0.14&0.25&0.89&0.16&0.19\\\\
>   0.13&0.13&0.15&0.20&0.92&0.23&0.13&0.15&0.20&0.92
>   \end{bmatrix}
> $$
>   Their DP matrices can be calculated as follows, with the circled positions in the boxes denoting the plan step and execution action matches:
> $$
> DP_A=\begin{bmatrix}
> 0.86&\boxed{0.82}&0.72&0.79&0.69&0.82&0.72\\\\
> 0.46&1.05&\boxed{1.68}&1.58&1.48&1.38&1.28\\\\
> 0.82&1.29&1.84&\boxed{2.63}&2.53&2.43&2.33\\\\
> 0.42&0.95&1.44&2.23&2.78&2.73&\boxed{3.35}
> \end{bmatrix},
> DP_B=\begin{bmatrix}
> 0.86&\boxed{0.82}&0.72&0.82&0.72&0.62&0.79&0.69&0.82&0.72\\\\
> 0.46&1.05&1.68&1.58&1.48&\boxed{1.48}&1.38&1.28&1.18&1.08\\\\
> 0.21&0.65&1.28&1.84&1.77&1.67&1.73&\boxed{2.27}&2.17&2.07\\\\
> 0.13&0.34&0.88&1.48&2.76&2.66&2.55&2.45&2.47&\boxed{3.09}
> \end{bmatrix}
> $$
>   We observe that the coordination score of trajectory A ($DP_A$=3.35/4=0.8375) is higher than that of trajectory B ($DP_B$=3.09/4=0.7725). This shows that the coordination score is sensitive to minor deviations; for instance, a single deviation like putting a unclean cloth at step $a^B_5$ leads to a decrease.
> For brevity, we have omitted some steps; the real drop from deviations is greater.
>
> > W1.3: Additional ablation studies could help clarify the contribution of different components in the framework.
>
> We conducted extensive ablation studies to analyze the contribution of each component, which cover three key aspects:
> 1. the core components for agent optimization (`Appendix A, Table 3, Page 12`);
> 2. the effect of the replanning method on data quality (`Table 2, Page 8`);
> 3. the efficacy of the coordination metrics (`Figure 4 and 5`).
>
> **Response to all Questions:** We sincerely appreciate the reviewer's attention to detail. We confirm that all raised issues have been corrected in the revised manuscript.
>
> **Response to all Limitations:** We observe that planning appears ineffective for short-horizon tasks, rather than a specific flaw of our approach. The table below shows the task performance of Qwen2.5-7B with/without explicit planning and the performance without explicit planning is even higher.
> |Dataset|w. Plan|Reward|
> |-|-|-|
> |HotPotQA|yes|28.25|
> |HotPotQA|no|34.92|
> |WebShop|yes|39.66|
> |WebShop|no|44.20|

---

> > ### Author Rebuttal · Reviewer_MqPe · 2026-04-03
> >
> > The authors of CoPE provide a response to the reviewers' concerns by leveraging concrete examples, empirical data, and a commitment to editorial precision.
> >
> > Regarding the framework's novelty, the authors acknowledge the use of established techniques like MCTS but emphasize that their core contribution lies in a novel coordination mechanism specifically designed to bridge the gap between planning and execution in long-horizon tasks. To demystify the coordination metric, they present an intuitive case study involving the task of placing a clean cloth in a drawer. By utilizing Dynamic Programming (DP) matrices, they demonstrate that the coordination score is highly sensitive to minor execution deviations, such as failing to clean the cloth before storage, which justifies the metric's effectiveness.
> >
> > The authors further substantiate their framework through extensive ablation studies detailed in the manuscript, which isolate the contributions of agent optimization components, replanning mechanisms, and the coordination metrics themselves. Addressing the limitation regarding short-horizon tasks, they provide comparative data from HotPotQA and WebShop, arguing that the diminished role of planning in these scenarios is an inherent characteristic of simpler tasks rather than a flaw in their methodology. Finally, they confirmed the rectification of all typographical and naming inconsistencies, such as the misspelled `{insturction}` variable and the "CoPlanExec" label, ensuring the clarity and reproducibility of the revised manuscript.

---

> > > ### Author Response · Authors · 2026-04-05
> > >
> > > We sincerely appreciate the reviewer’s accurate summary and positive assessment of our work. We are particularly encouraged by the recognition of our core contribution and the validity of our supporting evidence, including the case study, DP analysis, and ablation results. We are also glad that our clarifications on short-horizon benchmarks and textual corrections effectively addressed the concerns raised. The reviewer’s valuable feedback has significantly enhanced the clarity and quality of our manuscript.

---

### Official Review · Reviewer_c2iy · 2026-03-13

**Soundness:** 3
**Presentation:** 3
**Significance:** 3
**Originality:** 3
**Overall Recommendation:** 4
**Confidence:** 3

**Summary:**

This paper introduces CoPE, a framework for the joint optimization of planning and execution in LLM-based agents. The authors point out that existing methods overlook the inherent misalignment between long-term planning and multi-step execution—manifesting specifically as "Impractical Planning" and "Unfaithful Execution"—and that task rewards alone are insufficient to detect these issues.

**Compliance With Llm Reviewing Policy:**

Affirmed.

**Final Justification:**

The author addressed most of my concerns, and I believe the article met the acceptance criteria.

**Key Questions For Authors:**

see weakness

**Limitations:**

The authors do not appear to thoroughly discuss the limitations and potential negative societal impacts of their work, and it would be better to explicitly include such a discussion.

**Strengths And Weaknesses:**

Strengths
1. The article systematically analyzes the prevalence of "unrealistic planning" and "disloyal execution" among LLM agents, the motivation is clear.
2. The writing is good and clear.

Weaknesses
1. Self-Refining MCTS necessitates performing multiple rollouts at each node, invoking an LLM to self-rewrite failed trajectories, and subsequently computing a coordination metric based on multiple trajectories; the overall computational overhead can be substantial. The paper lacks an analysis of training costs, making it difficult to assess the framework's feasibility for larger-scale tasks.
2. Insufficient Diversity in Evaluation Benchmarks: The experiments utilize only two text-based environments with discrete action spaces: AlfWorld and ScienceWorld. In Appendix B, the authors acknowledge that for short-horizon tasks—such as WebShop and HotPotQA—the coordination gap approaches zero, implying significant limitations regarding the scenarios in which CoPE is applicable. The framework's efficacy remains unverified for environments with continuous action spaces, tool-calling scenarios (e.g., code-executing agents), or web-browsing tasks.
3. Questionable Generalizability of Coordination Metrics: The "Execution Compliance" metric relies on the compact embedding model `all-MiniLM-L6-v2` combined with rule-based matching; as this metric depends on a predefined action vocabulary (e.g., "goto," "take"), its feasibility is limited in scenarios featuring open or dynamic action spaces. Furthermore, the optimal values for the two penalty parameters, λ1 and λ2, vary across different environments (AlfWorld: 0.1/0.4; ScienceWorld: 0.2/0.4), necessitating per-environment tuning—a requirement that constrains the framework's "plug-and-play" capabilities.

---

> ### Author Rebuttal · Authors · 2026-03-29
>
> We sincerely thank the reviewer for valuable comments. We provide detailed responses and have incorporated the corresponding updates into our revised manuscript.
>
> > W1: The paper lacks an analysis of training costs, making it difficult to assess the framework's feasibility for larger-scale tasks.
>
> We appreciate the reviewer's concerns about the training efficiency. We provide an analysis based on theoretical complexity, empirical data, and key scalability factors:
>
> 1. **Theoretical Analysis.** The primary overhead comes from the evaluation part of `Self-Refining MCTS` with time complexity $T_{total}\approx O(M\cdot(n\cdot L_{exec}\cdot T_{LLM}))$, where $L_{exec}$ denotes the maximum step number and $T_{LLM}$ is the latency for a single LLM-environment interaction. This shows that as the scale increases, the cost grows linearly with respect to $M$ and $n$ rather than exponentially, guaranteeing controllable and predictable overhead.
>
> 2. **Practical Efficiency.** The table below shows the time consumption of each stage for a single agent optimization iteration with Qwen2.5-7B based on **four NVIDIA A100 GPUs**. We find: 1) The Data Collection has the highest overhead, but remains within practical scope; 2) The time cost of the Coordination Assessment is minimal.
>   |Dataset|Data Collection|Coordination Assessment|Agent Optimization|
>   |-|-|-|-|
>   |AlfWorld|2 day, 12 hours|30 minutes|2 hours|
>   |ScienceWorld|3 day, 3 hours |40 minutes|2 hours|
>
> 3. Feasibility is ensured by these factors: 1) **Offline Optimization:** The data can be stored in a replay buffer for repeated training; 2) **Parallelism in Evaluation:** The decoupled rollout allows fully parallel execution, making data collection highly concurrent on multi-GPU clusters; 3) **Agent Efficiency:** Table 1 demonstrates that CoPE agents have the shortest task completion trajectories, showing that the actual time is significantly lower than the theoretical upper bound of $L_{exec}$ interactions.
>
> In summary, CoPE’s computational overhead is theoretically linear and practically feasible for larger-scale tasks
>
> > W2: The framework's efficacy remains unverified for environments with continuous action spaces, tool-calling scenarios(e.g., code-executing agents), or web-browsing tasks.
>
> We appreciate the reviewer's constructive feedback regarding the evaluation scope. To address the concern about insufficient diversity in evaluation benchmarks, we conducted experiments on the challenging **2WikiMultihopQA** [1] benchmark, a multi-hop QA task that requires agents iteratively call a search engine tool with LLM-generated keywords to retrieve factual evidence, resulting in a highly flexible action space. We constructed 1000 samples as the training set and 100 samples as the testing set, and used answer exact match as the reward. As shown in the table below, CoPE achieves superior performance on 2WikiMultihopQA.
>
>   |Model|Method|Reward|
>   |-|-|-|
>   |GPT-4o|ReAct|57.31|
>   |Deepseek-Chat|ReAct|61.72|
>   |Gemini2.5-Pro|ReAct|51.60|
>   |Qwen2.5-7B|ReAct|40.37|
>   |Qwen2.5-7B|ETO|52.39|
>   |Qwen2.5-7B|RFT|49.76|
>   |Qwen2.5-7B|AGENTEVOL|54.35|
>   |Qwen2.5-7B|CoPE (Our)|**67.74** |
>
>   [1] Constructing a Multi‑hop QA Dataset for Comprehensive Evaluation of Reasoning Steps (COLING 2020)
>
> > W3.1: Questionable Generalizability of Coordination Metrics: ... its feasibility is limited in scenarios featuring open or dynamic action spaces.
>
> We appreciate the reviewer's concerns about the generalizability of metrics. Our coordination metrics have inherent generalizability and can operate effectively without dependence on specific scenarios, as evidenced by:
> 1. **Contextual Necessity:** The action vocabulary is provided by the task scenario itself, consistent with how LLM agents operate (requiring explicit API descriptions in the context window). This is not a limitation but a reflection of standard deployment constraints.
> 2. **Validation in Dynamic Spaces:** In our 2WikiMultihopQA experiments (featuring a dynamic action space), we disabled the rule-based matching component by setting α=0 in Algorithm 1. CoPE still outperformed other baselines relying solely on embedding-based similarity (`emb_cos`), confirming independence from specific datasets.
>
> > W3.2. Furthermore, the optimal values for the two penalty parameters, λ1 and λ2, vary across different environments (AlfWorld: 0.1/0.4; ScienceWorld: 0.2/0.4), necessitating per-environment tuning—a requirement that constrains the framework's "plug-and-play" capabilities.
>
> With regard to `λ1` and `λ2`, our hyperparameter analysis (`Appendix A, Figure 8, Page 11`) reveals an empirical pattern: setting $λ2/λ1>1$ consistently yields superior performance because aligning the planning step is more important. This guideline reduces the cost of hyperparameter tuning. Moreover, this analysis also confirms CoPE's effective plug-and-play capabilities, as it outperforms the 'w/o Coord-' baseline (`Figure 5, Page 8`) across all settings.

---

> > ### Author Rebuttal · Reviewer_c2iy · 2026-04-04
> >
> > Thank you for your reply; all my concerns have been addressed. I encourage the authors to add these new experiments to the final version of the paper, and I will maintain my positive grade.

---

> > > ### Author Response · Authors · 2026-04-05
> > >
> > > We sincerely appreciate the reviewer's constructive suggestions regarding the evaluation of generalization and computational overhead. We have incorporated all relevant discussions and experimental results into our revised manuscript.
> > > Your expert guidance was instrumental in significantly strengthening the rigor and clarity of our contributions.
> > >
> > > Given that you have kindly marked the concerns as "Fully resolved", we would be grateful if you could consider revisiting your current score to reflect this updated assessment.

---

### Decision · Program_Chairs · 2026-04-30

**Decision:**

Accept (regular)

**Comment:**

To bridge the gap where existing LLM optimization methods overlook the critical incoordination between planning and multi-step execution, this paper presents the CoPE framework that employs Self-Refining MCTS. It collects diverse plan-execution data, quantifies coordination via plan executability and execution adherence scores, and incorporates these scores as weights in coordination-weighted training objectives.

The reviewers acknowledged the clarity of motivation, the intuitive three-stage pipeline, and empirical improvements over baselines.
In the rebuttal, the authors satisfactorily addressed major concerns by providing additional experiments on 2WikiMultihopQA, as well as more analysis, training stability results across five seeds. These lead all four reviewers to mark their concerns as fully resolved and maintain or upgrade their scores (5, 4, 4, 4 scores). Given this consensus, I recommend the Acceptance decision.